# SpikingBrain: Spiking Brain-inspired Large Models

**Yuqi Pan**[1,2,3]**, Yupeng Feng**[1]**, Jinghao Zhuang**[1]**, Siyu Ding**[1]**, Han Xu**[1,4]**, Zehao Liu**[1,5]**,
**Bohan Sun**[1]**, Yuhong Chou**[1,5]**, Xuerui Qiu**[1,6]**, Anlin Deng**[1]**, Anjie Hu**[1,6,7]**, Shurong Wang**[1,8]**,
**Peng Zhou**[9]**, Man Yao**[1,2,3]**, Jibin Wu**[5]**, Jian Yang**[10]**, Guoliang Sun**[10]**, Bo Xu**[1,2*] **& Guoqi Li**[1,2,3*]

[1]Institute of Automation, Chinese Academy of Sciences
[2]Beijing Key Laboratory of Brain-Inspired General Intelligence Large Model
[3]Key Laboratory of Brain Cognition and Brain-inspired Intelligence Technology
[4]Beijing Academy of Artificial Intelligence    [5]The Hong Kong Polytechnic University
[6]Zhongguancun Academy    [7]Beihang University    [8]Zhejiang University
[9]LuxiTech    [10]MetaX Integrated Circuit Co., Ltd.

**Reviewed on OpenReview:** `https://openreview.net/forum?id=PNLShc0C6q`

## Abstract

Mainstream Transformer-based large language models (LLMs) face significant efficiency bottlenecks: training computation scales quadratically with sequence length, and inference memory grows linearly. These constraints limit their ability to process long sequences effectively. In addition, building large models on non-NVIDIA computing platforms poses major challenges in achieving stable and efficient training and deployment. To address these issues, we introduce SpikingBrain, a new family of brain-inspired models designed for efficient long-context training and inference. SpikingBrain leverages the MetaX[1] GPU cluster and focuses on three core aspects: i) **Model Architecture:** linear and hybrid-linear attention architectures with adaptive spiking neurons; ii) **Algorithmic Optimizations:** an efficient, conversion-based training pipeline compatible with existing LLMs, along with a dedicated spike coding framework; iii) **System Engineering:** customized training frameworks, operator libraries, and parallelism strategies tailored to the MetaX hardware. Using these techniques, we develop two models: **SpikingBrain-7B**, a linear LLM, and **SpikingBrain-76B**, a hybrid-linear MoE LLM. These models demonstrate the feasibility of large-scale LLM development on non-NVIDIA platforms, and our training framework supports weeks of stable training on hundreds of MetaX GPUs with Model FLOPs Utilization (MFU) at expected levels. SpikingBrain achieves performance comparable to open-source Transformer baselines while using exceptionally low data resources (continual pre-training of ∼150B tokens). Our models also significantly improve long-context efficiency and deliver inference with (partially) constant memory and event-driven spiking behavior. For example, SpikingBrain-7B achieves more than 100× speedup in Time to First Token (TTFT) for 4M-token sequences. Furthermore, the proposed spiking scheme achieves 69.15% sparsity, enabling low-power operation. Overall, this work demonstrates the potential of brain-inspired mechanisms to drive the next generation of efficient and scalable large model design. [2]

## 1 Introduction

Recent advances in large language models (LLMs) built on the Transformer architecture (Vaswani et al., 2017) have been driven by the scaling law (Kaplan et al., 2020), which suggests that performance improves with larger model sizes and more data (OpenAI, 2025; Google DeepMind, 2025; Anthropic, 2025). However,

---

[*]Corresponding authors: xubo@ia.ac.cn and guoqi.li@ia.ac.cn
[1]https://www.metax-tech.com/en
[2]The code for this project is publicly available at `https://github.com/BICLab/SpikingBrain-7B`.

this scale-driven approach comes with significant challenges: extremely high training costs, substantial energy consumption, and complex deployment pipelines. Therefore, achieving high performance and energy efficiency under limited resources has become a critical research goal. To address this, our work draws inspiration from brain mechanisms. We explore novel architectures, training paradigms, and spike coding schemes to develop efficient, brain-inspired LLMs that move beyond the traditional Transformer framework.

A further objective is to validate the training and deployment of such models on non-NVIDIA computing clusters. We use an open-source Transformer checkpoint (Qwen2.5-7B-base (Yang et al., 2024b) as an example) together with our efficient development framework to train and evaluate two models on the MetaX GPU cluster. The models, SpikingBrain-7B and SpikingBrain-76B-A12B, undergo end-to-end validation, including continual pre-training (CPT), long-context extension (up to 128k tokens), and supervised fine-tuning (SFT). We also adapt the vLLM inference framework to demonstrate deployment feasibility on MetaX hardware.

As illustrated in Figure 1, SpikingBrain constructs model architectures aligned with brain-inspired mechanisms; these architectures are developed via an efficient conversion pipeline and supported by adaptations for non-NVIDIA clusters. Our main technical contributions are as follows:

- **Hybrid Linear Architectures:** Moving away from quadratic self-attention, we design hybrid models combining linear, local, and standard attention modules. We explore two hybrid strategies: inter-layer sequential (SpikingBrain-7B) and intra-layer parallel (SpikingBrain-76B-A12B). The former achieves linear complexity and excels at long-context efficiency; the latter provides stronger language modeling capability through a more sophisticated architectural design. Notably, the modeling of linear attention (Katharopoulos et al., 2020) also resonates with neuronal dynamics in biological systems (He et al., 2024b).

- **Adaptive Threshold Spiking:** Inspired by event-driven biological neurons, we propose a spiking scheme that converts activations into integer spike counts and expands them into sparse spike trains. This enables addition-based, event-driven computation. Several encoding formats are supported, including binary $\{0, 1\}$, ternary $\{-1, 0, 1\}$, and bitwise spike coding. These sparse, event-driven representations provide the basis for our low-power inference design and may also inspire the development of next-generation neuromorphic hardware (Roy et al., 2019; Schuman et al., 2022; Frenkel et al., 2023).

- **Efficient Model Conversion:** i) **Attention modules:** Using a unified attention map analysis, we convert quadratic attention modules into sparse sliding-window and low-rank linear attention, by remapping the weights of existing Transformer models (Kasai et al., 2021). ii) **FFN modules:** For SpikingBrain-76B, we utilize an MoE upcycling technique (He et al., 2024a) that replicates dense FFN weights to create sparse experts, increasing capacity with minimal compute and memory overhead. These techniques, together with the dedicated spike coding framework, reduce continual training and inference costs, enabling efficient long-context handling with less than 2% of the compute required for training from scratch.

- **Large-Scale Training and Inference on MetaX:** Both models are trained on hundreds of MetaX C550 GPUs, covering the entire pipeline, from data preprocessing to distributed training and inference. We adapt frameworks, operators, and communication primitives to ensure stability. This work represents, to our knowledge, the first large-scale training of brain-inspired LLMs on a non-NVIDIA platform, achieving stable training at 76B parameters.

Our design philosophy holds that linear attention modules exhibit modeling characteristics highly analogous to human memory mechanisms, particularly through their use of compressed, continuously updated memory states (Gu, 2025). In addition, the Mixture-of-Experts (MoE) component reflects a principle of modular specialization, akin to the distributed and specialized processing observed in biological neural networks (O'Doherty et al., 2021). By combining network-level sparsity (via MoE) with neuron-level spiking sparsity, our approach offers a robust multi-scale efficiency strategy. Taken together, these results point toward a promising direction for developing more efficient and biologically plausible large-model architectures that extend beyond standard Transformer-based LLMs through brain-inspired mechanisms.

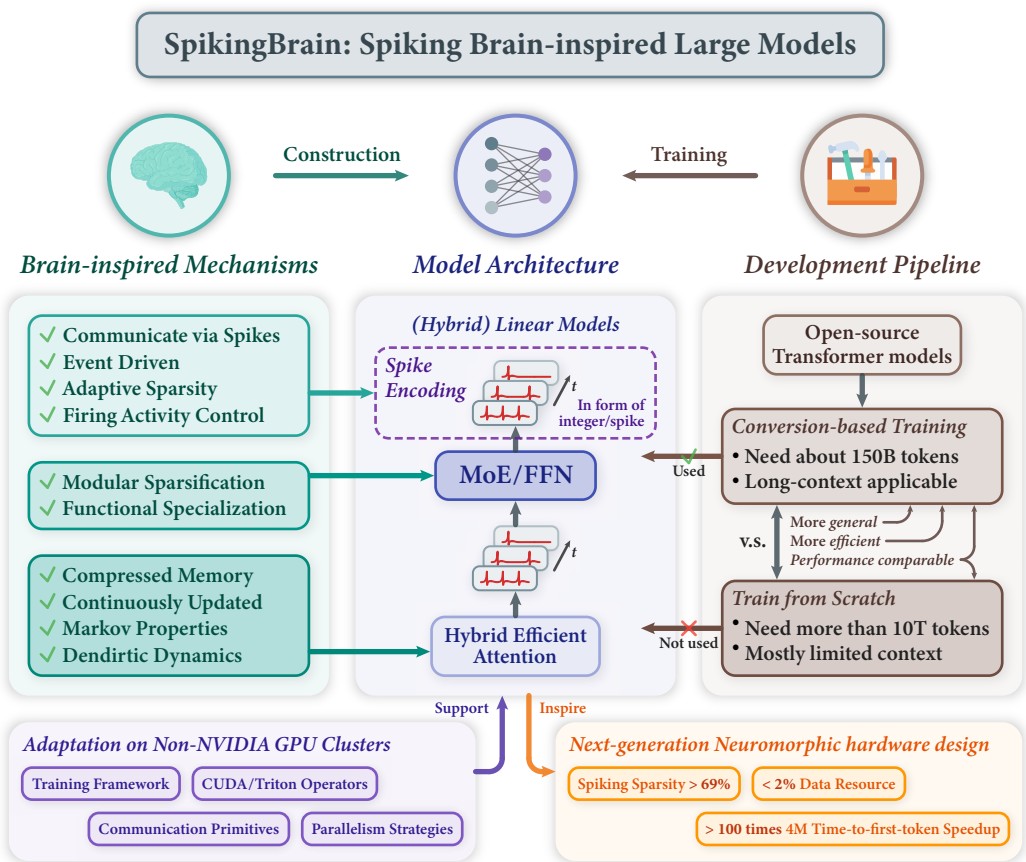

Figure 1: **Overview of SpikingBrain.** Inspired by brain mechanisms, SpikingBrain integrates hybrid efficient attention, MoE modules, and spike encoding into its architecture, supported by a universal conversion pipeline compatible with the open-source model ecosystem. This enables continual pre-training with less than 2% of the data while achieving performance comparable to mainstream open-source models. We further adapt frameworks, operators, parallel strategies, and communication primitives for non-NVIDIA (MetaX) clusters, ensuring stable large-scale training and inference. SpikingBrain achieves over $100\times$ speedup in TTFT for 4M-token sequences, while spiking delivers over 69% sparsity at the micro level. Combined with macro-level MoE sparsity, these advances provide valuable guidance for the design of next-generation neuromorphic chips.

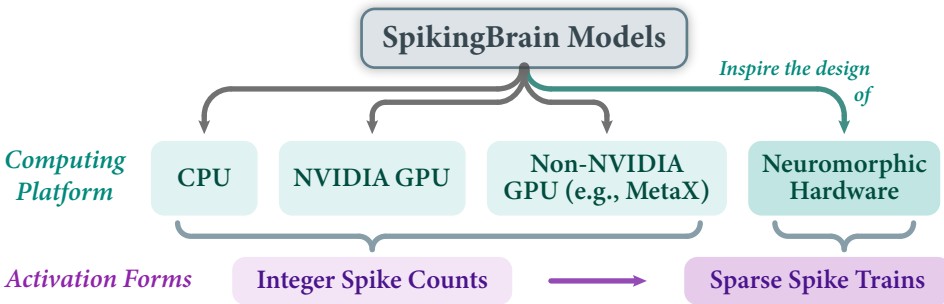

Figure 2: **Compatibility of SpikingBrain models across diverse computing platforms.** SpikingBrain models can be deployed on CPUs and both NVIDIA and non-NVIDIA GPUs using integer activation formats, also inspiring the design of neuromorphic hardware leveraging event-driven sparse spike representations.

In summary, this work presents efficient brain-inspired LLM development on MetaX GPUs, producing two models: SpikingBrain-7B and SpikingBrain-76B MoE. They achieve (near-)linear complexity, high training stability, and competitive performance through continual pre-training, requiring only $\sim 2\%$ of the data typically needed for training comparable LLMs from scratch. Inference shows substantial speedups, with SpikingBrain-7B reaching over $100\times$ speedup in TTFT (Time to First Token) for 4M-token inputs. We further deploy a compressed 1B SpikingBrain model on a CPU-based mobile inference framework, achieving a $15.39\times$ speedup at a sequence length of 256k. The proposed spiking scheme also yields $\sim 69\%$ sparsity, offering strong support for reducing power consumption. These findings highlight the potential of brain-inspired design for future neuromorphic hardware and enable efficient model scaling across diverse computing platforms (see Figure 2).

## 2 Model Architecture

### 2.1 Core Components

#### 2.1.1 Attention Mechanisms

The core temporal interaction module of an LLM is the attention mechanism, which projects input tokens into QKV vectors. At each time step, the current query $\mathbf{q}_t$ interacts with past keys and values $\mathbf{k}_s, \mathbf{v}_s (s \leq t)$ to produce the output $\mathbf{o}_t$.

**Softmax Attention**  Standard softmax attention (Vaswani et al., 2017) captures global, token-wise interactions across the entire sequence:

$$\mathbf{O} = \text{softmax}(\mathbf{Q}\mathbf{K}^\top \odot \mathbf{M})\mathbf{V}; \quad \mathbf{o}_t = \frac{\sum_{s=1}^t \exp(\mathbf{q}_t\mathbf{k}_s^\top)\mathbf{v}_s}{\sum_{s=1}^t \exp(\mathbf{q}_t\mathbf{k}_s^\top)}. \tag{1}$$

Here, $\mathbf{M}$ is a causal mask where $\mathbf{M}_{ij} = 1$ if $i \geq j$ and $\mathbf{M}_{ij} = -\infty$ if $i < j$. For a sequence of length $n$, softmax attention provides high arithmetic intensity and parallelism during training but incurs $\mathcal{O}(n^2)$ computation. During inference, maintaining the key-value cache adds an $\mathcal{O}(n)$ memory footprint, becoming a bottleneck for long-context tasks.

**Sliding Window Attention (SWA)**  To reduce this quadratic computational cost, many linear-complexity variants have been proposed. One example is SWA (Beltagy et al., 2020; Child et al., 2019; Jiang et al., 2023), which limits the attention scope to a fixed local window of size $w$, capturing fine-grained local interactions:

$$\mathbf{O} = \text{softmax}(\mathbf{Q}\mathbf{K}^\top \odot \mathbf{M}')\mathbf{V}; \quad \mathbf{o}_t = \frac{\sum_{s=t-w+1}^t \exp(\mathbf{q}_t\mathbf{k}_s^\top)\mathbf{v}_s}{\sum_{s=t-w+1}^t \exp(\mathbf{q}_t\mathbf{k}_s^\top)}. \tag{2}$$

Here, $\mathbf{M}'$ is a windowed causal mask where $\mathbf{M}'_{ij} = 1$ if $i - w + 1 \leq j \leq i$, otherwise $\mathbf{M}'_{ij} = -\infty$. Thus, training complexity and inference memory reduce to $\mathcal{O}(n)$ and $\mathcal{O}(1)$ respectively, as $w$ is a fixed constant independent of $n$.

**Linear Attention**  Another common variant is linear attention (Katharopoulos et al., 2020; Yang et al., 2024c; Dao & Gu, 2024), which removes the softmax function to achieve linear complexity and can be reformulated as a state-based linear recurrence:

$$\mathbf{O} = (\mathbf{Q}\mathbf{K}^\top \odot \mathbf{M})\mathbf{V}; \tag{3}$$

$$\mathbf{o}_t = \sum_{s=1}^t (\mathbf{q}_t\mathbf{k}_s^\top)\mathbf{v}_s = \mathbf{q}_t \sum_{s=1}^t (\mathbf{k}_s^\top\mathbf{v}_s) = \mathbf{q}_t\mathbf{S}_t, \text{ where } \mathbf{S}_t = \mathbf{S}_{t-1} + \mathbf{k}_t^\top\mathbf{v}_t. \tag{4}$$

Here, $\mathbf{M}$ is a causal mask where $\mathbf{M}_{ij} = 1$ if $i \geq j$ and $\mathbf{M}_{ij} = 0$ if $i < j$. Linear attention supports chunk-wise parallelism during training and maintains a fixed-size state in recurrent form (Sun et al., 2023; Yang et al., 2024c), reducing computational complexity and inference memory to $\mathcal{O}(n)$ and $\mathcal{O}(1)$.

**Hybrid Attention**   Each attention mechanism has distinct strengths: softmax attention excels at global retrieval, SWA is efficient for local context, and linear attention effectively compresses long-range information. Hybrid Attention aims to combine these methods to balance efficiency and accuracy. Two common paradigms are: i) **Inter-layer sequential hybridization** (Lieber et al., 2024; Ren et al., 2024; De et al., 2024; Blakeman et al., 2025; Qwen Team, 2025; Team et al., 2025), where different attention types are stacked across layers:

$$\mathbf{o}_t^1 = \text{attention1}(\mathbf{x}_t), \quad \mathbf{o}_t = \text{attention2}(\mathbf{o}_t^1). \tag{5}$$

ii) **Intra-layer parallel hybridization** (Dong et al., 2024; Zuo et al., 2025), where different attention modules process the same input in parallel and outputs are merged:

$$\mathbf{o}_t^1 = \text{attention1}(\mathbf{x}_t), \quad \mathbf{o}_t^2 = \text{attention2}(\mathbf{x}_t), \quad \mathbf{o}_t = w_1 \cdot \mathbf{o}_t^1 + w_2 \cdot \mathbf{o}_t^2. \tag{6}$$

Both approaches are widely adopted, enabling flexible trade-offs between modeling accuracy and efficiency by adjusting the proportions of each attention type. Our SpikingBrain-7B employs inter-layer hybridization of linear attention and SWA, whereas SpikingBrain-76B uses intra-layer hybridization that combines linear attention, SWA, and full softmax attention.

### 2.1.2   Mixture-of-Experts (MoE)

The core idea of sparse MoE (Lepikhin et al., 2020; Fedus et al., 2022) is to enhance model capacity by introducing $N$ parallel expert networks into the original feed-forward network (FFN) layer, while dynamically selecting the most relevant $k$ experts for each input token $\mathbf{x}$ through a router $\mathbf{W}_r$:

$$\mathbf{p} = \sigma(\mathbf{W}_r \mathbf{x}), \quad \mathcal{I} = \{i \mid p_i \in \text{top-}k(\mathbf{p})\}, \tag{7}$$

$$\text{MoE\_activation}(\mathbf{x}) = \sum_{i \in \mathcal{I}} p_i \times E_i(\mathbf{x}). \tag{8}$$

Here, each expert $E_i(\cdot)$ is an FFN. The router produces a probability vector $\mathbf{p}$, where $\sigma$ is typically a softmax or sigmoid function. For each token, only the top-$k$ experts with the highest probabilities are activated (with $\mathcal{I}$ denoting the set of activated experts). The final MoE output is then computed as the weighted sum of these selected experts.

Unlike traditional dense layers, MoE does not activate all experts simultaneously. Instead, it leverages sparse activation to significantly increase parameter capacity and expressiveness while keeping computational cost nearly unchanged. This property enables MoE to combine efficiency with high performance in both training and inference. Prior studies (Dai et al., 2024; Yang et al., 2024a; Liu et al., 2024) further suggest that introducing shared experts that are always active, and retaining a few dense FFN layers in shallow stages of the model can improve training stability and overall performance.

In practice, dense models can be efficiently expanded into MoE models using the **upcycling** technique (Komatsuzaki et al., 2022; He et al., 2024a), which scales parameter capacity without sacrificing the original performance. The procedure involves: i) replicating the dense FFN weights across all experts so that the upcycled model initially matches the dense baseline; ii) rescaling expert outputs appropriately to maintain consistency with the output scale of the original model.

### 2.1.3   Spiking Neuron Modeling

Spiking neurons are the core components of Spiking Neural Networks (SNNs) (Maass, 1997), and their modeling generally falls into two paradigms: simplified approximations and detailed simulations. The **Hodgkin–Huxley (HH)** model (Hodgkin & Huxley, 1952) represents the latter, accurately describing the membrane potential dynamics of biological neurons through a set of nonlinear differential equations. It focuses on simulating the synergistic effects of transmembrane ion fluxes. Although such models can faithfully replicate fine-scale bioelectrical phenomena of neurons (e.g., complex firing patterns and refractory periods), they entail high computational complexity due to the involvement of multiple high-order differential operations. Thus, HH-based models are currently mainly suitable for physiological studies on small neuron populations and cannot meet the efficiency and latency requirements of LLMs.

By contrast, simplified models such as the **Leaky Integrate-and-Fire (LIF)** model, are more commonly used in practice (Yao et al., 2023; 2024a). The LIF neuron is a first-order approximation of soma dynamics incorporating the temporal dimension. For an input token $\mathbf{x}$ (extended over time, where $\mathbf{x}_t$ is the input at the $t$-th time step), the membrane potential $\mathbf{v}_t$ and spike output $\mathbf{s}_t$ of the neuron can be formulated as below:

$$\mathbf{v}_{t+1} = \begin{cases} \lambda(1-\mathbf{s}_t)\mathbf{v}_t + \mathbf{v}_{\text{reset}} \cdot \mathbf{s}_t + \mathbf{x}_{t+1} & \text{hard reset} \\ \lambda\mathbf{v}_t - V_{\text{th}} \cdot \mathbf{s}_t + \mathbf{x}_{t+1} & \text{soft reset} \end{cases} ; \qquad \mathbf{s}_t = 1 \; \text{if} \; \mathbf{v}_t \geq V_{\text{th}} \; \text{else} \; 0. \tag{9}$$

Here, the membrane potential accumulates charge in the soma, $\lambda$ represents a decay factor (mimicking the natural leakage of ion potential), where $V_{\text{th}}$ represents a fixed firing threshold, and $\mathbf{v}_{\text{reset}}$ is the reset potential, which is usually set to 0. While this model simplifies the ion-based mechanisms and retains the core logic of natural neurons, it still exhibits several limitations for large-scale models (Luo et al., 2025; Yao et al., 2025; Fan et al., 2025): i) The temporal dynamics (mainly introduced by the decay factor and reset mechanism) lead to increased training complexity and instability; ii) Even when integrated into pre-trained models, redundant complexity still persists, making it complicated to build models and simulate the original values; iii) The fixed threshold can cause suboptimal spike generation, leading to many neurons becoming silent or over-activated, posing challenges for optimization and impeding the balance between accuracy and energy efficiency.

To address these limitations, we propose **Adaptive-threshold Spiking Neurons**, which simplify LIF neurons for enhanced computational efficiency and modeling accuracy, while drawing inspiration from adaptive neuronal thresholding:

- **Adaptive Threshold**: Inspired by adaptive dynamic properties of biological neurons, the firing threshold $V_{\text{th}}$ is designed as a dynamic value correlated with the membrane potential. This prevents neurons from being over-excited or over-quiescent, maintaining a moderately active state from a statistical perspective.

- **Simplified Temporal Computation**: The decay factor is eliminated, and the soft-reset mechanism is adopted, enabling the conversion from continuous values to integer spike counts in a single step. During optimization, the temporal dimension is merged for stability and computational efficiency on GPUs. For inference, the temporal dimension can be re-expanded, and energy-efficient computation can be achieved using sparse event-driven asynchronous hardware (Yao et al., 2024b).

Our modeling preserves essential characteristics of biological neurons and is appropriately simplified to eliminate redundant computations. It not only leverages the energy efficiency advantages of biological neural systems but also holds promise for effective engineering implementation in pre-trained LLMs when combined with specific asynchronous hardware.

## 2.2 Integrated Model Architectures

Through lightweight training, a base Transformer can be converted into efficient attention variants of different forms, alongside MoE upcycling and spiking neuron coding (see Section 3). As a case study, we develop two models from the Qwen2.5-7B-base checkpoint: **SpikingBrain-7B**, a pure linear model optimized for long-context efficiency, and **SpikingBrain-76B**, a hybrid-linear MoE model designed to balance efficiency and performance. Their architectures are illustrated in Figure 3. Both models are integrated with HuggingFace and vLLM inference frameworks, supporting deployment on single- and multi-GPU settings. Additionally, as shown in Figure 2, SpikingBrain models can be deployed across diverse computing platforms, including CPUs as well as both NVIDIA and non-NVIDIA GPUs. Moreover, by expanding integer activations into equivalent sparse spike representations, the models provide insights that can inspire the design of next-generation neuromorphic chips.

**SpikingBrain-7B** The 7B model achieves purely linear complexity by interleaving linear attention and sliding window attention (SWA) layers with a fixed 4K window in a 1:1 ratio. The FFN modules adopt the same SwiGLU design as the base model. Spike coding is applied to convert the activations of all linear

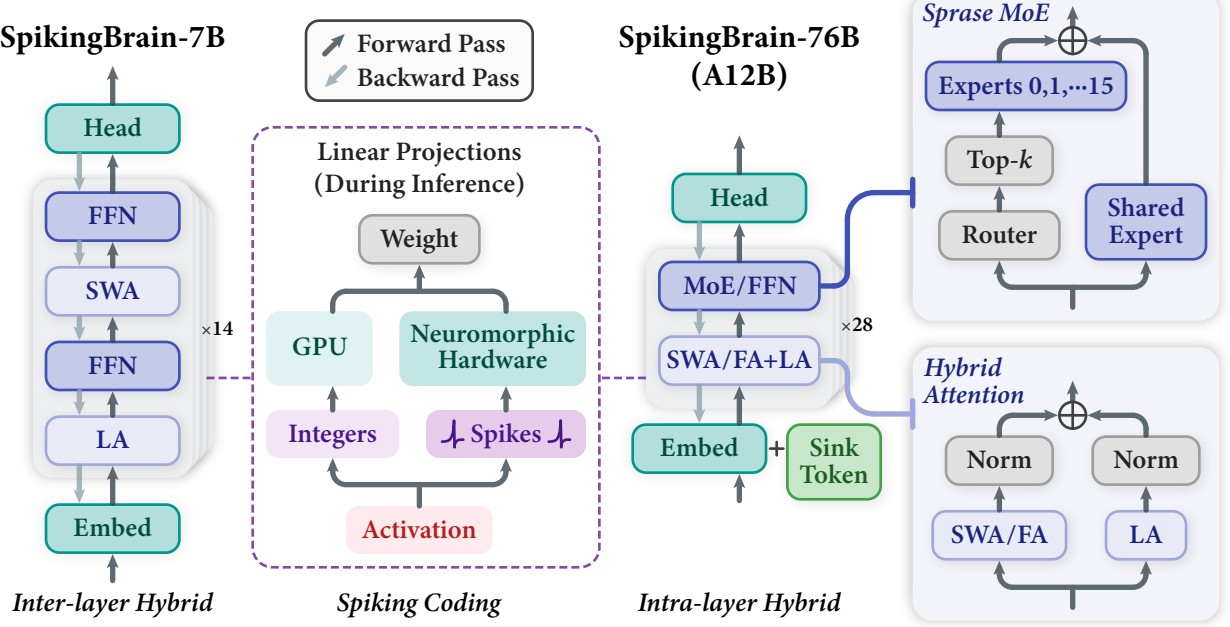

Figure 3: **Integrated architectures of SpikingBrain models.** FA: Full Softmax Attention; SWA: Sliding Window Attention; LA: Linear Attention. *(Left)* SpikingBrain-7B is a linear model with inter-layer hybridization. *(Middle)* Spike coding converts activations into integer counts for GPU execution or into spike trains for event-driven neuromorphic hardware. *(Right)* SpikingBrain-76B is a hybrid-linear MoE model with intra-layer hybridization, configured with 128 sink tokens, 16 routed experts, and 1 shared expert. Seven dense FFNs are located at layers $[1, 2, 3, 5, 7, 9, 11]$, with all other FFNs implemented as MoE layers. Attention modules are arranged as "LA + FA" at layers $[7, 14, 21, 28]$, and "LA + SWA" at all other layers.

projection layers in the model, as matrix multiplications for parameter projections dominate the model's FLOPs. In this architecture, the inter-layer hybrid design is optimized for extreme efficiency, owing to its simplicity in attention computation, model parallelism and sequence parallelism. SWA layers captures precise local patterns, while linear attention layers efficiently compresses long-range information. The model thus provides linear-time training complexity and constant memory usage during inference, regardless of sequence length, offering substantial efficiency gains for long-context processing. In practice, we implement linear attention using a Gated Linear Attention Module Yang et al. (2024c) (GLA), where the gating vector $\mathbf{g}_t$, derived from low-rank projection parameters, enhances expressivity and recurrent modeling capability:

$$\mathbf{S}_t = \mathrm{diag}(\mathbf{g}_t) \odot \mathbf{S}_{t-1} + \mathbf{k}_t^\top \mathbf{v}_t, \tag{10}$$

$$\mathbf{o}_t = \mathbf{q}_t \mathbf{S}_t. \tag{11}$$

**SpikingBrain-76B**   The 76B model integrates linear attention and SWA in a 1:1 intra-layer parallel hybrid configuration, while standard full-attention layers are interleaved at a 1:6 ratio across layers. This intra-layer hybrid design is optimized for stronger language modeling capabilities, as it enables more intricate attention computation and incorporates additional attention parameters and FLOPs. During parallel hybridization, outputs of both attention branches are normalized to ensure consistent scale and to prevent instability in the early stages of training. In addition, 128 learnable sink tokens (Xiao et al., 2023; Dong et al., 2024) are introduced to mitigate the attention-sink phenomenon in softmax attention and to enhance the flexibility of SWA for local modeling. Specifically, 128 trainable embeddings are prepended to the input; these tokens are attended by all other tokens and also attend to each other without causal masking. This functionality is implemented by customizing FlashAttention (Dao, 2024) kernels. The model also employs Gated Linear Attention Modules, but in this case, the gating vector is tied directly to the key vector (Chou et al., 2024; Qin et al., 2024), i.e., $\mathbf{g}_t = \mathbf{1} - \mathbf{k}_t$, eliminating the need for extra gating parameters. The FFN modules adopt

a sparse Mixture-of-Experts (MoE) design: each MoE layer consists of 16 routed experts (top-1 activated) and one shared expert, so that only about 15% of parameters (12B) are activated per token. To stabilize training and control parameter growth, 7 dense FFN layers are preserved out of 28 total layers (specifically at layers 1, 2, 3, 5, 7, 9, and 11). Similarly, spike coding is applied to the activations of all linear projection layers throughout the model.

**Connections with Brain Mechanisms**  Our architectural choices are closely aligned with principles observed in biological brains. i) Linear attention modules exhibit modeling properties analogous to human memory, relying on compressed and continuously updated states (Gu, 2025). At each step, they retrieve information only from the current memory, showing Markov-like behavior. From a biological perspective, their stateful temporal recurrence can be viewed as an abstract computational analogy to neuronal dendritic dynamics (Wang et al., 2025). ii) The Mixture-of-Experts (MoE) component embodies the principle of modular sparse activation and functional specialization, reminiscent of the distributed and specialized processing found in neural circuits (O'Doherty et al., 2021). iii) Our spike coding scheme draws inspiration from event-driven and adaptively sparse neuronal activation in biological systems (Maass, 1997). By combining network-level sparsity (MoE) with neuron-level spiking sparsity, our approach enables on-demand allocation of computation and provides a robust two-scale efficiency strategy. Collectively, these results suggest a promising pathway for designing large-model architectures that are both efficient and biologically plausible (Li et al., 2024).

## 3 Training Paradigm

The development of our SpikingBrain models relies on a highly efficient and universal conversion pipeline that is compatible with the open-source model ecosystem. Specifically, we start from a Transformer base model and perform targeted transformations on three key modules—hybrid attention modules, MoE modules, and spike-coding modules—supported by multi-stage training (including CPT and SFT) to adapt the model to the new brain-inspired architectures.

### 3.1 Attention Map Correspondence

Here, we provide a brief analysis of the relationships among different attention maps to illustrate the feasibility of transferring parameters across attention types. First, the attention map in pre-trained Transformers is defined as:

$$\mathbf{A} = \text{softmax}(\mathbf{Q}\mathbf{K}^\top \odot \mathbf{M}) \in \mathcal{R}^{n \times n}. \tag{12}$$

On this basis, SWA can be interpreted as a sparsified version of this map with a strong recency bias (Beltagy et al., 2020; Child et al., 2019; Zaheer et al., 2020):

$$\mathbf{A} = \text{softmax}(\mathbf{Q}\mathbf{K}^\top \odot \mathbf{M}'). \tag{13}$$

Moreover, linear attention can be regarded as a low-rank approximation of the attention map, where the maximum rank of $\mathbf{A}$ is $d$ (the dimensionality of the query/key vectors) (Chen et al., 2021; Wang et al., 2020):

$$\mathbf{A} = \mathbf{Q}\mathbf{K}^\top \odot \mathbf{M}. \tag{14}$$

Leveraging this attention-map correspondence, we can initialize the QKV projection parameters of SpikingBrain directly from a pre-trained Transformer checkpoint (Kasai et al., 2021), and this applies to full attention, linear attention, and SWA. By reusing the learned QK similarity and performing lightweight training on a small amount of data, the attention can be adapted to these local or low-rank cases. Since SWA maintains local modeling precision and linear attention retains global interaction, combining them in a hybrid paradigm provides a closer approximation to the original attention map (Dass et al., 2023; Chen et al., 2021; Arora et al., 2024b). This enables a smoother transition during conversion and accelerates loss convergence.

Finally, due to the generality of this attention-map formulation, we can construct diverse attention patterns—sparse, local, linear, or hybrid—and transfer them from any pre-trained softmax attention model. This

allows for flexible trade-offs between performance and efficiency by adjusting factors such as the proportion of retained full-attention layers or the window size of local attention.

To ensure stable convergence during the conversion stage, we follow several key practices:

- **Apply non-negative activation to the QK vectors in linear attention** (Kasai et al., 2021; Mercat et al., 2024), such as ReLU in SpikingBrain-7B or Sigmoid in SpikingBrain-76B. Because softmax attention maps are inherently non-negative, the converted attention must preserve this property to enable smooth transfer. Normalization, as in softmax attention, is applied after the linear attention outputs.

- **Keep newly introduced parameters low-rank**, such as normalization layers, gating components and sink tokens. In conversion training, the learning rate is typically lower and the dataset smaller than in pre-training, making it difficult to optimize a large number of randomly initialized parameters. We also want the pre-trained weights to guide optimization; therefore, we reuse all projection weights in the attention and FFN modules and minimize new parameters.

- **Perform long-context extension during conversion**. Since efficient attention mechanisms scale sub-quadratically and are substantially more efficient on long contexts than the base Transformer, a practical approach is to restrict the training length and resource usage of the original quadratic model, and then integrate long-context extension with continual pre-training during conversion. This allows increasing the context length while maintaining training efficiency, as shown in the conversion pipeline of SpikingBrain (Section 3.4).

- **Fully train the model during conversion to ensure performance**, using learning rate warm-up and either cross-architecture distillation (Wang et al., 2024; Zhang et al., 2024b) or full-parameter training (Zhang et al., 2024c). For simplicity and to fit the memory capacity of MetaX GPUs, we adopt full-parameter training without freezing the backbone.

## 3.2 MoE Upcycling

To efficiently expand a dense model into an MoE architecture, we adopt the upcycling technique (Komatsuzaki et al., 2022), which increases model capacity while reusing the knowledge encoded in the original parameters. At initialization, the FFN in the base dense model is replicated into $N$ experts, and a randomized router is introduced. The router selects the top-$k$ experts for each token with probability $\mathbf{p}$ and outputs their weighted sum, ensuring functional equivalence to the original dense FFN at the start:

$$E_1(\mathbf{x}) := E_2(\mathbf{x}) := \cdots := E_N(\mathbf{x}) := \text{FFN}(\mathbf{x}), \quad \text{at initialization.} \tag{15}$$

$$\mathbf{p} = \sigma(\mathbf{W}_r\mathbf{x}), \quad \mathcal{I} = \{i \mid p_i \in \text{top-}k(\mathbf{p})\}, \tag{16}$$

$$\text{MoE\_activation}(\mathbf{x}) = \sum_{i \in \mathcal{I}} p_i \times E_i(\mathbf{x}). \tag{17}$$

During training, stochastic routing and data noise gradually break the symmetry among experts, leading to differentiated gradients and specialized expert weights. In addition to the $N$ routed experts, our 76B model includes $S$ shared experts ($S = 1$ in our setting) that are always active for all tokens, helping stabilize the conversion process.

Directly replicating and activating multiple experts, however, amplifies the output scale. To maintain consistency between MoE and dense FFN outputs, we rescale the expert weights (He et al., 2024a). The relationship between MoE activation $a_{\text{MoE}}$ and dense activation $a_{\text{dense}}$ are derived as below, and the scaling factor $\alpha$ can be defined accordingly:

$$a_{\text{MoE}} = S \times a_{\text{dense}} + \frac{k}{N} \times a_{\text{dense}} = \left(S + \frac{k}{N}\right) a_{\text{dense}}, \qquad \alpha = \sqrt[3]{\frac{1}{S + \frac{k}{N}}}. \tag{18}$$

We apply this factor to both shared and routed experts at initialization, so that $E_i(\mathbf{x}) := \alpha \times \text{FFN}$ holds for all $1 \le i \le N$ at initialization. For SpikingBrain-76B, we use $S = 1, k = 1, N = 16$, yielding $\alpha = 0.98$.

### 3.3 Spiking Driven LLMs

Inspired by biological computation mechanisms (event-driven and sparse activation) and aiming to balance performance with efficiency, we propose a dedicated spiking strategy that encodes activations as equivalent integer values and spike sequences. This method can be applied both during and after training, converting the activations of large models into spikes. To further improve energy efficiency, we quantize both model weights and the KV cache to INT8 precision in conjunction with the spiking process. Integrated with SpikingBrain's lightweight conversion pipeline, this approach requires no full fine-tuning; a small calibration set is sufficient to optimize quantization parameters. For SpikingBrain-7B, the entire optimization process takes about 1.5 hours on a single GPU with 15 GB memory, significantly reducing deployment cost while preserving accuracy.

Our activation spiking scheme follows a decoupled two-step approach: i) **Adaptive-threshold spiking during optimization**: single-step generation of integer spike counts while maintaining appropriate neuronal firing activity. ii) **Spike coding during inference**: expansion of spike counts into sparse spike trains over virtual time steps. This approach enables the integer-based formulation to support computationally efficient optimization on GPUs, while the expanded spiking formulation provides event-driven, energy-efficient inference when combined with specialized hardware.

#### 3.3.1 Step 1: Adaptive Threshold Spiking

The first step, adaptive-threshold spiking, focuses on the single-step generation of integer spike counts. At this stage, activations are converted using a simplified adaptive-threshold neuron model. The core idea is to design a dynamic threshold that ensures neurons remain statistically balanced—neither over-excited or over-quiescent—thereby avoiding redundant spikes or information loss often caused by fixed thresholds. Specifically, we define:

$$\mathbf{x} = \sum_{t=1}^{T} \mathbf{x}_t, \quad \mathbf{s}_{\text{INT}} = \sum_{t=1}^{T} \mathbf{s}_t, \quad V_{\text{th}}(\mathbf{x}) = \frac{1}{k} \operatorname{mean}(\operatorname{abs}(\mathbf{x})), \tag{19}$$

where $\mathbf{x}$ denotes the continuous floating-point activations at the projection layer of the large model, equivalent to the accumulated multi-step inputs $\mathbf{x}_t$ in conventional SNNs. $V_{\text{th}}(\mathbf{x})$ is the adaptive threshold determined by the membrane potential, $k$ is a hyperparameter that controls the firing rate, and $\mathbf{s}_{\text{INT}}$ is the aggregated integer spike count.

We adopt simplified adaptive-threshold spiking neurons to convert continuous activations into integer spike counts. By eliminating the decay factor (i.e., using an IF neuron model) and utilizing the soft-reset mechanism, the inference dynamics can be expressed as:

$$\mathbf{v}_{t+1} = \mathbf{v}_t - V_{\text{th}} \cdot \mathbf{s}_t + \mathbf{x}_{t+1}, \qquad \mathbf{s}_t = 1 \ \text{ if } \ \mathbf{v}_t \geq V_{\text{th}}(\mathbf{x}) \ \text{ else } \ 0. \tag{20}$$

During optimization, the temporal dimension can be collapsed into a single-step computation:

$$\mathbf{v}_T = \mathbf{x}, \quad \mathbf{s}_{\text{INT}} = \operatorname{round}\left(\frac{\mathbf{v}_T}{V_{\text{th}}(\mathbf{x})}\right). \tag{21}$$

This single-step integer formulation enables computationally efficient and stable optimization on GPUs, while adaptive threshold maintains appropriate firing rates. As a result, our spiking scheme preserves the essential characteristics of biological neurons, yet is sufficiently simplified to be practical for large-scale models.

For gradient computation in the optimization phase, since the spiking process is an approximate representation of the original membrane potential, the gradient of the spiking process can be skipped (i.e., using straight-through estimation), which maintains the gradient computation process consistent with conventional training.

Furthermore, controlling firing activity is the primary motivation for adopting adaptive-threshold neurons rather than LIF neurons in our SpikingBrain architecture. The effects of the adaptive threshold and hyperparameter $k$ on firing activity can be summarized as follows:

- **Threshold dynamics**: $V_{\text{th}}(\mathbf{x})$ is determined by the mean absolute membrane potential. When the input potential is high, $V_{\text{th}}(\mathbf{x})$ increases, preventing excessive spiking and thus controlling sparsity to

reduce redundant computation. When the input potential is low, $V_{\mathrm{th}}(\mathbf{x})$ decreases, allowing neurons to emit a small number of spikes to retain key information and avoid accuracy loss from inactivity. Statistically, membrane potentials often follow a long-tailed Gaussian-like distribution with occasional outliers. In this setting, the mean absolute value approximates 0.8 times the standard deviation, providing a stable metric for regulating spike activity.

- **Effect of the hyperparameter** $k$: $k$ directly scales the threshold and thereby determines the distribution of spike counts. A larger $k$ lowers $V_{\mathrm{th}}(\mathbf{x})$, producing higher spike counts $\mathbf{s}_{\mathrm{INT}}$ and broader firing ranges. This is suited for accuracy-critical scenarios where additional computation is acceptable. A smaller $k$ raises $V_{\mathrm{th}}(\mathbf{x})$, reducing spike counts and producing sparser encodings, which is advantageous for low-power edge deployments. Tuning $k$ thus provides a flexible trade-off between accuracy and efficiency. This trade-off is empirically evaluated using SpikingBrain-7B (Appendix A.8).

- **Outlier handling**: Large models often contain rare but high-magnitude activations (outliers) that significantly affect accuracy. The adaptive threshold, due to its statistical basis, is robust to these values. Neurons maintain stable activity for the majority of inputs while producing higher spike counts for rare outliers. This behavior resembles the burst response of biological neurons, ensuring that critical information carried by outliers is preserved. On specialized asynchronous hardware, these rare bursts have minimal impact on overall energy efficiency.

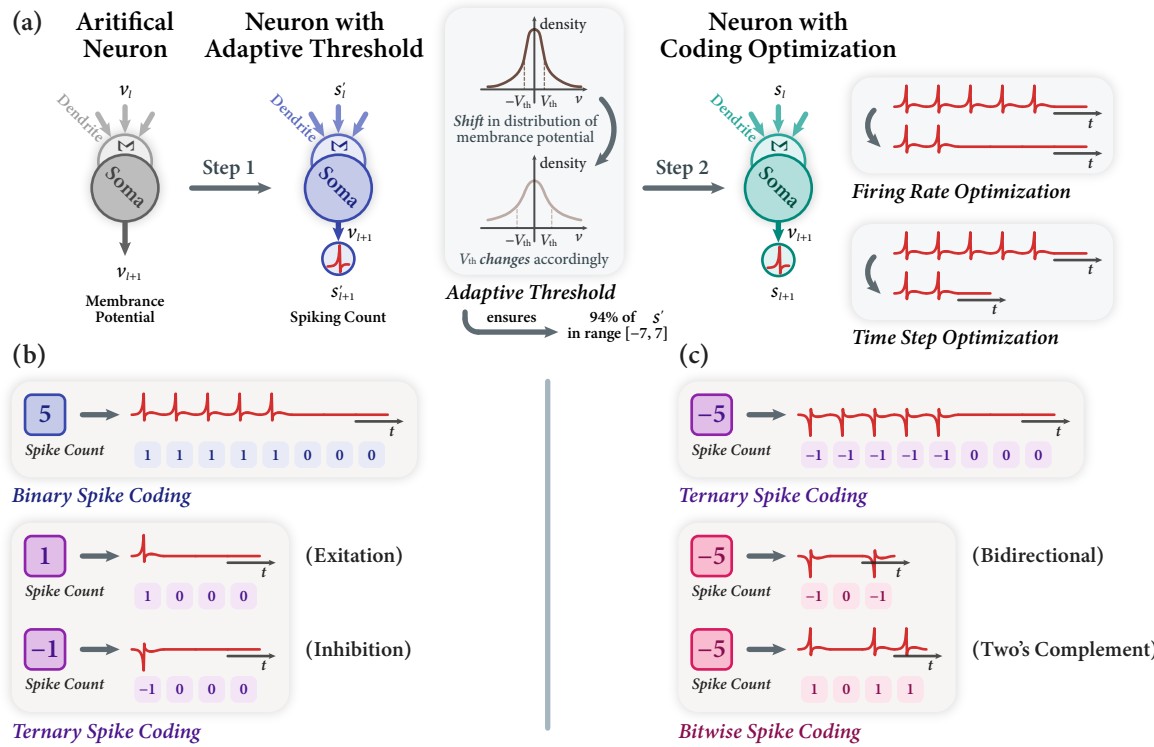

Figure 4: **Schematic of three spike coding schemes.** (a) An adaptive threshold maps membrane potential to spike counts, which are expanded over virtual timesteps into sparse spike trains, enabling the conversion from continuous activations to discrete spikes. (b) Ternary vs. Binary: binary uses $\{0,1\}$ to represent "spike/no-spike", while ternary uses $\{-1,0,1\}$ to encode both excitatory and inhibitory events. Compared with binary, ternary reduces both timesteps and firing rate by half. (c) Bitwise vs. Ternary: spike counts are unfolded into binary bits across timesteps to achieve temporal compression. In high-count scenarios, the required timesteps are far fewer than ternary, leading to significantly higher efficiency.

### 3.3.2 Step 2: Spike Coding

During inference, the integer spike count $\mathbf{s}_{\text{INT}}$ generated in Step 1 is expanded into a sparse spike sequence with values $\{-1, 0, 1\}$ along the time dimension to support event-driven computation, which formulated as:

$$\mathbf{s}_{\text{INT}} = \sum_{t=1}^{T} \mathbf{s}_t, \quad \mathbf{y} = V_{\text{th}}(\mathbf{x}) \cdot \sum_{t=1}^{T} \mathbf{W}\mathbf{s}_t, \tag{22}$$

where $\mathbf{W}$ is the weight matrix and $\mathbf{y}$ is the output of the linear projection layer. Because each $\mathbf{s}_t$ takes values from $\{0, 1, -1\}$, matrix multiplications are replaced by event-driven accumulations, thereby improving computational efficiency.

To expand $\mathbf{s}_{\text{INT}}$ into multi-step spikes $\mathbf{s}_t$, we design three encoding schemes tailored to different application needs. The primary goal is to replace dense matrix multiplications with sparse, event-driven additions when supported by appropriate hardware, while optimizing the spike firing rate and number of time steps without sacrificing representational capacity. This reduces power consumption and increases computational sparsity.

- **Binary Spike Coding** $\{0, 1\}$: This is the most basic event-driven spike coding method. Each time a unit spike is fired (with a value of 1), it represents an activation of the neuron state, and the spike count is accumulated over continuous time steps. This coding scheme is intuitive and simple, with low computational overhead, making it suitable for scenarios with very low spike counts and effectively reducing system complexity. However, when representing large counts, this coding often requires many time steps to complete the representation. Additionally, the lack of directional optimization for spike firing results in a higher firing rate, further limiting system energy efficiency. Meanwhile, this encoding scheme only supports positive integers (unsigned $\mathbf{s}_{\text{INT}}$) which is not applicable to Adaptive Threshold Spiking.

- **Ternary Spike Coding** $\{-1, 0, 1\}$: To improve neuronal expressivity and sparsity in event-driven computations, we introduce ternary spike coding. The core idea is to extend the traditional binary spike coding by adding inhibitory spikes $(-1)$, resulting in three firing states: $\{-1, 0, 1\}$. Compared to binary coding, which can only represent positive values, ternary coding offers bidirectional expression capabilities. Here, "1" represents excitatory spikes, "$-1$" inhibitory spikes, and "0" the silent state, making it more aligned with the "excitation/inhibition" regulatory mechanism in biological neural systems. As shown in Figure 4 (b), ternary spike coding not only provides symmetric expression capabilities of $\pm 1$ but also reconstructs the mapping from membrane potential to spike counts through a symmetric quantization strategy. This strategy maps activations to a symmetric distribution $[-k, \ldots, 0, \ldots, +k]$ rather than $[0, 1, 2, \ldots]$, so that low-amplitude counts (e.g., $\pm 1$) statistically occupy a larger probability mass, effectively absorbing high-frequency large counts from the tail of the original distribution. As a result, this scheme halves the number of time steps and reduces the spike firing rate by more than $2\times$, significantly improving sparsity and energy efficiency without sacrificing expressivity.

- **Bitwise Spike Coding**: Bitwise coding can be viewed as an event sequence encoding method, where integer count values are expanded bit by bit into spike events over time steps. Each time step corresponds to one binary bit of the count value, significantly compressing the time dimension overhead. As shown in Figure 4 (c), this mechanism supports three implementation forms to accommodate different symbolic representations and precision requirements: i) **Pure bitwise encoding**, suitable for positive integers, provides extremely high temporal compression in high-count scenarios. For instance, a count of 256 requires 256 consecutive time steps in binary coding, 128 in ternary coding, but only 8 steps in 8-bit bitwise encoding. ii) **Bidirectional bitwise encoding** uses $\pm 1$ to represent each bit, replacing negative part with $-1$. This halves the time step emission rate and reduces the required number of time steps by one (e.g., 7 steps for a count of 256), while maintaining equivalence in representation. iii) **Two's complement encoding** incorporates sign information into the highest bit, supporting both positive and negative counts while retaining binary simplicity. Compared to ternary coding, which still requires many time steps for higher bit counts, the bitwise encoding scheme significantly compresses time steps, reducing total spike emissions by up to $8\times$ (for an 8-bit

count). This effectively reduces overall spike communication overhead and computational load while maintaining precision, making it particularly suitable for high-precision, low-power, and time-sensitive neuromorphic computing tasks. Taking Bidirectional bitwise encoding as an example, its encoding formula and the equivalent computation of matrix multiplication can be expressed as:

$$\mathbf{s}_{\text{INT}} = \sum_{t=1}^{T} 2^{t-1}\mathbf{s}_t, \quad \mathbf{y} = V_{\text{th}}(\mathbf{x}) \cdot \sum_{t=1}^{T} 2^{t-1}\mathbf{W}\mathbf{s}_t, \tag{23}$$

### 3.3.3   Hardware Adaptation and Deployment Potential

Our spiking scheme can be executed on GPUs. By collapsing the temporal dimension into a single step, it avoids the incompatibility between event-driven computation and the synchronous architecture of GPUs, enabling direct simulation and inference validation on general-purpose hardware.

However, the synchronous design of GPUs cannot fully exploit the event-driven and sparse-asynchronous advantages of spiking signals (Schuman et al., 2022; Frenkel et al., 2023). GPUs operate under fixed high-frequency clock cycles, unlike biological neural systems that remain idle without spikes and trigger computation only upon activation. To take full advantage of our design and leverage the low-power potential, implementing matrix operations on specialized asychornous hardware architectures (e.g., neuromorphic chips, or spiking processors) is required. These platforms natively respond to sparse spike events without requiring clock synchronization (Roy et al., 2019; Yao et al., 2024b): circuits remain in a quiescent, low-power state in the absence of spikes and perform addition operations only when spikes occur.

This approach maximizes energy efficiency and offers a practical pathway for deploying low-power brain-inspired LLMs in edge scenarios such as industrial control and mobile devices. It also outlines a reference technology roadmap for developing next-generation energy-efficient neuromorphic hardware, supporting the shift of large-scale models from compute-driven to energy-optimized paradigm. Additional discussion of hardware implementation considerations is provided in Appendix C.

### 3.4   Multi-stage Conversion Pipeline

The multi-stage conversion pipeline consists of continual pre-training (CPT) with long-context extension, followed by supervised fine-tuning (SFT). For demonstration purposes, all training data of SpikingBrain are sampled from high-quality open-source datasets; in practice, domain-specific data can be incorporated for vertical adaptation.

The continual pre-training process comprises three stages which progressively extend the context window while adapting the model to new architectures. In the first stage, our models are trained on 100B tokens with a sequence length of 8k, aiming to transfer attention patterns toward local or low-rank variants and ensure loss convergence. Subsequently, the second and third stages extend the sequence length to 32k and 128k, respectively, each trained with 20B to 30B tokens. The entire conversion process consumes about 150B tokens in total. Compared to the ∼10T tokens typically required for training from scratch, the CPT approach requires only about 2% of the data, enabling efficient adaptation under resource and budget constraints. All three stages use the Matrix (Zhang et al., 2024a) dataset, with long-context data generated through a simple per-domain packing strategy. The RoPE base remains consistent with the base model at 1M.

SFT is conducted in three stages, each employing domain-specific data to progressively enhance the capabilities of the model in general knowledge, dialogues, and reasoning. The first stage focuses primarily on improving fundamental language understanding and domain knowledge, using the Infinity Instruct foundational (Li et al., 2025) dataset covering various foundational topics such as scientific knowledge, code interpretation, and mathematical problem solving. This stage is trained with 500k samples under an 8k sequence length. The second stage specializes in dialogue ability and instruction following, utilizing the Infinity Instruct chat (Li et al., 2025) dataset that includes multi-turn conversations, task-oriented dialogues, and knowledge-based question answering. The data volume and sequence length remain consistent with the first stage. The third stage targets reasoning tasks, using a high-quality reasoning dataset (Liu et al., 2025; Madhusudhan et al., 2025) distilled via the DeepSeek-R1 (Guo et al., 2025) method, containing examples with detailed

chain-of-thought annotations for mathematical proofs, logical reasoning, case analyses, and other multi-step inference problems. To ensure cross-linguistic reasoning, the dataset maintains a 1:1 Chinese-English ratio. A total of 150k samples are used under a sequence length of 8k.

# 4 Implementation on the MetaX Cluster

Distributed training of brain-inspired large models on the MetaX non-NVIDIA cluster poses several challenges. These include: ensuring stability for large-scale parallel training, sustaining intensive communication under long-sequence parallel topologies, and adapting CUDA and Triton operators for hybrid attention. In this work, we introduce targeted optimizations to address each of these challenges, which enable the successful training of both the SpikingBrain-7B and -76B models. This section details the adaptations for distributed training, operator customization, and the parallel topologies used in our conversion pipeline.

## 4.1 Distributed Training Adaptation

To enable efficient and stable training on non-NVIDIA GPU clusters, the MetaX software platform has implemented a series of hardware-aware adaptations across multiple dimensions, including MoE optimization, computation-communication overlap, memory optimization, auto-tuning, distributed cluster training, and kernel fusion. Some of these optimizations are designed as plug-in modules, enhancing the flexibility of the training framework.

For MoE training, four strategies are introduced to mitigate memory and computational pressure during the early training stages:

- **Hot-Cold Expert Optimization**: To address communication hotspots caused by uneven token routing in the early phases, frequently accessed experts are replicated locally. This reduces communication overhead until expert load stabilizes, after which the local copies are removed.

- **Adaptive Recomputation**: When a heavily utilized expert processes tokens beyond a set threshold, activation recomputation is triggered to save memory. This technique is automatically disabled in later stages when the token distribution is balanced.

- **Multi-Granularity Recomputation** (Korthikanti et al., 2022): To balance computation and memory under high memory pressure, experts support three recomputation levels: lightweight (activations and router), moderate (including FFN and shared experts), and full (entire MoE layer).

- **Length Alignment**: Variations in token counts per expert can affect the efficiency of GEMM. Token dropping and padding are applied to unify input sequence lengths, improving overall computational efficiency.

To address high communication overhead in distributed training, MetaX utilizes SDMA engines for intra-node high-speed data transfer. For tensor parallelism and expert parallelism, communication kernels are fused with compute kernels to reduce scheduling conflicts (NVIDIA, 2025). Memory optimizations include fine-grained offloading of transformer layer weights or optimizer states to the CPU, as well as selective recomputation of normalization layers, activation functions, or individual transformer layers based on memory demand.

For long-sequence parallelism, the MetaX GPU cluster supports efficient multi-GPU and multi-node communication with stable training throughput. This alleviates synchronization challenges and improves both GPU memory utilization and compute efficiency. Inter-GPU connectivity is enhanced through MetaLink and PCIe 5.0, eliminating intra-node bottlenecks. RDMA over InfiniBand 200/400G or RoCE ensures low-latency, high-bandwidth inter-node communication, meeting the requirements of large-scale distributed training.

The software stack also incorporates auto-tuning and fast recovery mechanisms for large-scale clusters:

- **Auto-Tuning**: An automated tuning engine covers operators, memory, and communication. It benchmarks common operators, models communication performance across network topologies, and explores parallel configuration spaces to recommend top-k strategies, minimizing manual effort.

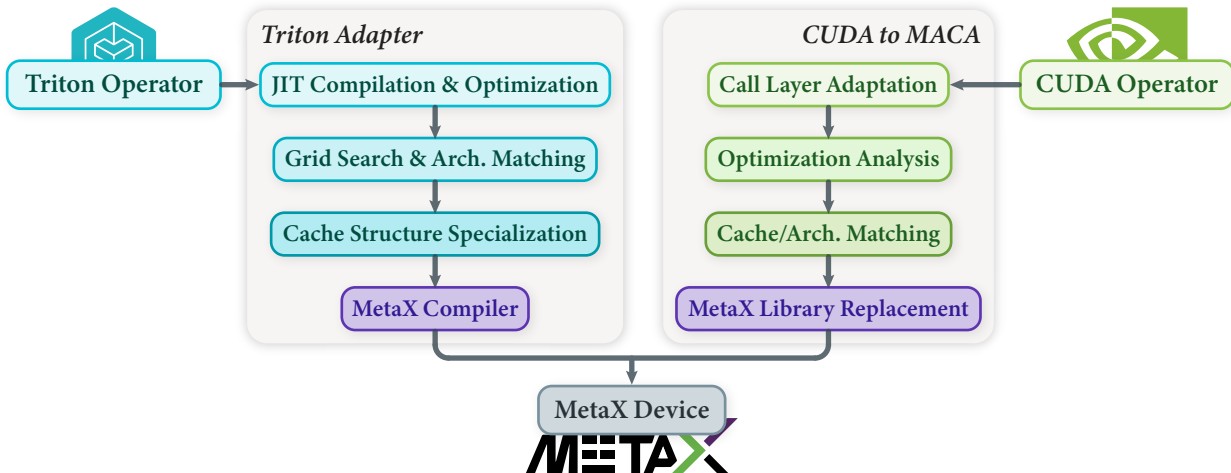

Figure 5: **Operator adaptation of SpikingBrain on MetaX GPUs.** The adaptation involves two complementary pathways: Triton adaptation and CUDA migration to MACA framework, covering different operator subsets. Together, they form a unified hardware adaptation framework tailored for MetaX GPUs.

- **Fast Checkpointing**: The DLRover (Wang et al., 2023) Flash Checkpoint technique first writes training state (model weights, optimizer, learning rate scheduler, etc.) to CPU memory before asynchronously persisting to a distributed file system. This reduces I/O time by 85% and shortens recovery time after failures. The built-in profiling tools automatically instrument training jobs, monitor performance per layer, detect slow nodes, and trigger alerts to sustain high cluster utilization.

### 4.2 Operator Adaptation

The efficient adaptation of SpikingBrain large models to MetaX GPUs relies on the comprehensive MetaX software ecosystem. The overall process can be divided into two major components: Triton adaptation and CUDA migration to the MetaX self-developed MACA framework. These two pathways target different operator subsets in SpikingBrain models. While distinct in implementation, they complement each other and together constitute a complete hardware adaptation framework tailored for MetaX GPUs (see Figure 5).

**In the Triton adaptation workflow**, we designed four progressive stages based on the MetaX technology stack and Triton's compilation pipeline. The objective is to fully exploit the MetaX GPU's strengths in compilation optimization and instruction scheduling, while keeping the adaptation process transparent to higher-level applications:

- **JIT Compilation Optimization**: During Triton's just-in-time compilation stage, the linear attention operators (Yang et al., 2024c) used in SpikingBrain are reorganized at the code level. By refining instruction pipelining and register allocation, the operators achieve a balance between memory access latency and computational density. This stage directly leverages Triton's hardware-agnostic general optimization strategies, enabling dynamic and efficient execution of compute kernels.

- **Grid Search and Architecture Matching**: Systematic exploration of Block/Grid configurations is carried out and mapped to the scale of MetaX GPU streaming multiprocessors and thread concurrency features. By extending Triton's architectural support to the MetaX GPU family, we significantly improve the throughput of matrix multiplication in linear attention operators.

- **Cache Structure Specification**: To minimize redundant computation and memory traffic, we introduce fixed, structured cache designs aligned with MetaX's on-chip memory hierarchy. For instance, by optimizing reuse strategies for weight matrices and Key-Value sequences in line with

L2 cache capacity and bandwidth properties, inference efficiency for long-sequence processing is markedly improved.

- **Target Code Generation via MetaX Compiler**: Finally, Triton kernels are transformed by the MetaX compiler backend into executable target machine code for MetaX GPUs. Beyond standard code generation, the compiler introduces deep optimizations for tensor cores, SIMD instruction sets, and memory alignment constraints, ensuring operators run at near-peak hardware performance.

**In the CUDA-to-MACA migration workflow**, four tightly integrated stages are involved:

- **Call Layer Adaptation**: Original CUDA APIs and runtime interfaces are encapsulated and redirected to the MACA framework, allowing seamless mapping without altering user-facing code. This significantly reduces migration overhead for developers.

- **Optimization Analysis**: Using performance profiling tools, we identify performance bottlenecks in our SpikingBrain models, such as long-sequence attention operators involving softmax, exp/sum, dot-product accumulation, and certain normalization operations. These operators are reimplemented using native MACA optimizations to fully leverage tensor acceleration units.

- **Cache and Architecture Matching**: Similar to the Triton adaptation process, cache-awareness is embedded into the MACA execution engine. Key data structures (e.g., intermediate accumulation matrices, positional encoding caches) are retained in high-speed cache, reducing global memory traffic and improving efficiency through hardware-specific optimizations.

- **Replacement with MetaX Acceleration Libraries**: Finally, core baseline operators are substituted with MetaX's high-performance libraries. The Hotspot operator acceleration library is used to replace foundational calls, while advanced libraries such as mcFlashInfer2, specifically optimized for MetaX hardware, are employed to deliver sustained performance improvements at the hardware level.

It is worth emphasizing that the entire adaptation and migration process for brain-inspired large models on MetaX GPUs adheres to a user-friendly design philosophy. Whether through Triton kernel optimization or CUDA-to-MACA migration, end users can preserve their existing programming practices and interface calls. In most cases, no extensive code modifications are required for the models to run efficiently on MetaX GPUs. Meanwhile, the MetaX ecosystem provides unified debugging and performance analysis tools, enabling developers to transparently monitor execution characteristics on hardware and tune performance as needed.

### 4.3 Parallel Topology

The memory demands of training large language models often exceed the capacity of a single GPU. To make training such models feasible, it is essential to employ efficient and scalable distributed training techniques alongside other memory reduction methods. These approaches distribute computational and storage loads while maintaining training efficiency. In this section, we present the four parallelization strategies and corresponding distributed training topologies employed to train SpikingBrain models on the MetaX cluster.

**Data Parallelism (DP)** Data parallelism (Valiant, 1990) involves partitioning the training data into batches, each processed by a separate GPU. Each GPU maintains a complete replica of the model and performs forward and backward passes independently, while gradient synchronization occurs only at the beginning of the backward propagation. Data parallelism offers low communication overhead and excellent scalability. As computing resources increase, the scale of data parallelism can grow accordingly, increasing the total batch size to enhance overall throughput and reduce training time. During training, we employ ZeRO (Rajbhandari et al., 2020) to eliminate redundancy of the optimizer states, effectively distributing the GPU memory pressure.

**Pipeline Parallelism (PP)** Pipeline parallelism divides the model into different stages by layers. Each GPU is assigned a subset of layers, and intermediate activations are communicated between GPUs during the forward and backward passes using peer-to-peer (p2p) communication. We employ the 1F1B (Narayanan et al., 2021; Harlap et al., 2018) scheduling algorithm for efficient pipeline execution. Furthermore, incorporating Mixture-of-Experts (MoE) layers interspersed with dense FFN layers in SpikingBrain-76B helps mitigate the memory imbalance issue, where the first pipeline stage typically exhibits higher memory usage compared to other stages.

**Expert Parallelism (EP)** In Mixture-of-Experts models, most parameters reside within the expert networks. Expert parallelism (Rajbhandari et al., 2022) addresses this by partitioning experts across multiple GPUs. This requires all-to-all communication to dispatch and combine tokens. During the training of SpikingBrain-76B, we employ Grouped GEMM kernels to accelerate computation across experts, and an auxiliary loss is used to promote balanced token routing across experts.

**Sequence Parallelism (SP)** Training with long sequences incurs substantial memory costs for storing intermediate activations. Sequence parallelism tackles this by splitting the input sequence along its length dimension, with each GPU processes a contiguous chunk. Communication primitives like all-to-all (a2a) or peer-to-peer (p2p) are then used to exchange necessary information between devices, enabling the computation of attention over the entire sequence. For our experiments, we employ DeepSpeed Ulysses (Jacobs et al., 2023) for efficient sequence parallelism and incorporate specialized algorithms for linear attention modules to further improve performance. For linear attention, AllGather communication is used to collect intermediate local states (Sun et al., 2025) when the SP size is small, while for larger SP sizes (e.g., across multiple nodes), the All-Scan primitive from ZeCO (Chou et al., 2025) sequence parallelism is adopted to effectively eliminate communication overhead.

**Parallel Topology of SpikingBrain** The SpikingBrain-7B model is trained using the Colossal-AI (Li et al., 2023b) framework. For 128k sequence length training, a parallel strategy of 32-way data parallelism and 8-way sequence parallelism is adopted. To reduce GPU memory consumption, techniques such as ZeRO-2 (Rajbhandari et al., 2020) and activation recomputation (Chen et al., 2016; Korthikanti et al., 2022) are applied. Sequence parallelism is implemented with DeepSpeed Ulysses (Jacobs et al., 2023), which performs all-to-all communication within nodes. In addition, the number of attention heads is padded to satisfy the partition requirements.

The SpikingBrain-76B model is trained using the Megatron (Shoeybi et al., 2019) framework. For 8k sequence length training, the parallel configuration consists of 128-way data parallelism, 8-way expert parallelism, and 4-way pipeline parallelism. ZeRO (Rajbhandari et al., 2020) and selective activation recomputation (Chen et al., 2016; Korthikanti et al., 2022) are applied to reduce memory consumption. For extended contexts of 32k and 128k, 4-way and 8-way sequence parallelism are added on top of the base 8k parallel strategy, respectively. The 76B hybrid model integrates linear attention and (local) softmax attention. The softmax attention branch uses DeepSpeed Ulysses (Jacobs et al., 2023) with all-to-all communication and head padding to enable partitioning, while the linear attention branch employs the AllGather-based SP algorithm (LASP-2 (Sun et al., 2025)) to reduce communication overhead.

## 5 Results

### 5.1 Downstream Performance

We conduct a comprehensive evaluation of the two converted efficient models, SpikingBrain-7B and -76B, across a broad set of downstream tasks[3]. They are compared against open-source baselines of similar scale, including Hybrid, Transformer, and MoE architectures, as well as the strong base model Qwen2.5-7B. To ensure fairness, all evaluations are performed under identical settings using the OpenCompass (Buitrago & Nystrom, 2019) framework. In selecting evaluation metrics, we place greater emphasis on pretraining-oriented

---

[3]This section evaluates the effectiveness of the architectural design and conversion pipeline. All models remain in floating-point format; results for spiking models are presented in Section 5.5.

benchmarks (e.g., MMLU, C-Eval), as these better indicate whether our models—trained with fewer than 200B tokens during efficient conversion—can inherit the knowledge and modeling capacity of the base model. Moreover, these benchmarks are less confounded by data quality factors such as alignment performance after SFT. Note that, except for Qwen2.5, the other baselines are trained on limited Chinese data, resulting in disadvantages on CMMLU and C-Eval. For more information on the baselines and benchmarks, please refer to Appendix A.

**Pre-trained Models**  As shown in Table 1, our SpikingBrain-7B linear model recovers nearly 90% of the base model's performance across benchmarks, reaching a level comparable to advanced Transformer models such as Mistral-7B and Llama3-8B. This indicates that, with appropriate training strategies, linear attention architectures can preserve strong general-purpose modeling ability while substantially reducing complexity. However, due to the significant structural modifications of the attention mechanism, there remains a certain performance gap between the 7B pure linear model and base Qwen2.5-7B. This suggests that the extent of recovery is strongly influenced by architectural changes when pursuing extreme efficiency.

Table 1: **Performance evaluation of the SpikingBrain-7B pre-trained model.** All models are tested with the HuggingFace framework and evaluated using a perplexity-based method.

| | SpikingBrain-7B | Falcon-Mamba | Mistral | Zamba-v1 | Llama3.1 | Qwen2.5 (base) |
|---|---|---|---|---|---|---|
| Params | 7B | 7B | 7B | 7B | 8B | 7B |
| Tokens | +150B | 5.8T | – | 1T | 15T | 18T |
| Complexity Type | Linear | Linear | Linear | Hybrid | Quadratic | Quadratic |
| **Benchmarks** | | | | | | |
| MMLU | 65.84 | 63.24 | 62.56 | 58.19 | 65.74 | **74.21** |
| CMMLU | 71.58 | 42.50 | 44.58 | 38.42 | 52.44 | **81.73** |
| ARC-C | 43.32 | 47.53 | 45.13 | 37.18 | **51.96** | 44.04 |
| HellaSwag | 70.89 | 71.50 | **75.81** | 52.02 | 71.60 | 72.81 |
| Ceval | 69.80 | 41.93 | 47.04 | 36.40 | 51.46 | **81.60** |

As shown in Table 2, our SpikingBrain-76B hybrid-linear MoE model nearly closes the performance gap with its base model by scaling parameter size and adopting more fine-grained attention designs. Despite using fewer activated parameters, it achieves performance comparable to, and in some cases surpassing, representative Transformer models such as Llama2-70B, Mixtral-8×7B, and Gemma2-27B. Compared with SpikingBrain-7B, the 76B model delivers consistent improvements across all benchmarks, providing strong evidence that within the conversion framework, flexible choices in attention and FFN design allow a better balance between performance and efficiency, while enabling architectures tailored to specific application needs.

Table 2: **Performance evaluation of the SpikingBrain-76B pre-trained model.** All models are tested with the vLLM (Kwon et al., 2023) framework and evaluated using a perplexity-based method.

| | SpikingBrain-76B (76B-A12B) | Jamba (52B-A12B) | Mixtral-8*7B (47B-A13B) | LLama2-70b | Gemma2-27B | Qwen2.5 (base) |
|---|---|---|---|---|---|---|
| Params | 12B/76B | 12B/52B | 13B/47B | 70B | 27B | 7B |
| Tokens | +160B | – | – | 2T | 13T | 18T |
| Complexity Type | Hybrid | Hybrid | Quadratic | Quadratic | Quadratic | Quadratic |
| **Benchmarks** | | | | | | |
| MMLU | 73.58 | 67.17 | 71.23 | 69.57 | **75.94** | 75.31 |
| CMMLU | 78.83 | 51.11 | 52.70 | 52.94 | 61.80 | **81.50** |
| ARC-C | 42.00 | 49.10 | 48.98 | 46.21 | **57.64** | 43.56 |
| HellaSwag | 73.31 | 79.39 | **79.42** | 79.15 | 79.34 | 73.37 |
| Ceval | 78.89 | 48.94 | 54.39 | 49.26 | 61.02 | **81.68** |

**Instruct Models**  We conduct a three-stage SFT process on the converted pre-trained models to endow them with instruction-following and chain-of-thought reasoning abilities. As shown in Table 4, our aligned

models achieve performance on par with open-source chat baselines of similar scale across general knowledge, long-sequence modeling, and instruction following. Importantly, the SFT process does not cause overfitting or degrade the general capabilities acquired during pretraining, indicating that the (hybrid) linear architectures maintain both stability and scalability under alignment training. Given current data limitations, we primarily aim to demonstrate effective alignment results obtained on MetaX computing clusters, leaving large-scale and higher-quality post-training alignment as future work.

**Long-context Capability**  To assess real-world long-context modeling and understanding, we evaluate SpikingBrain-76B (after stage-2 SFT) on six LongBench tasks (Bai et al., 2024). Given the baseline's maximum positional embeddings, we truncate prompts to 32k. As shown in Table 5, our model is competitive with other hybrid and Transformer baselines while using only four global softmax attention layers. Additionally, LongBench evaluates long-context modeling and understanding rather than recall performance, where linear attention exhibits known limitations. This limitation and its empirical analysis are discussed in Appendix A.4.

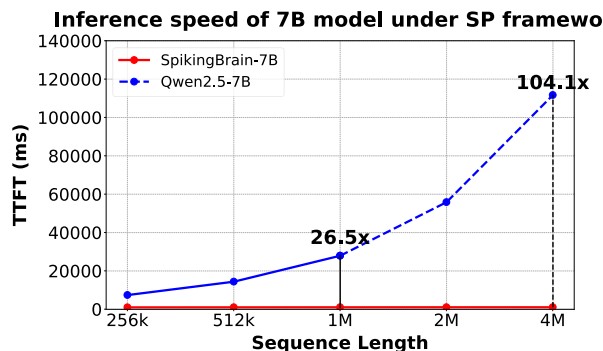
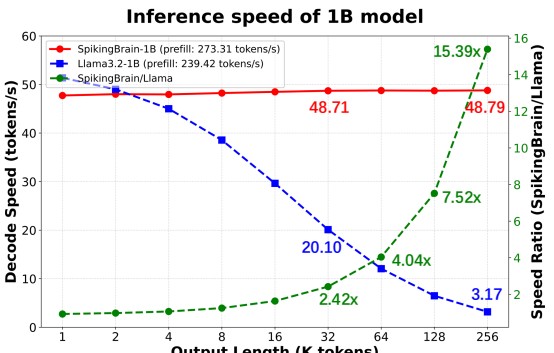

Figure 6: **TTFT comparison under sequence parallelism.** Time to First Token (TTFT) latency of SpikingBrain-7B compared with the Qwen2.5-7B baseline across different input lengths, measured on NVIDIA H100 GPUs. For inputs beyond 2M tokens, direct evaluation of Qwen2.5-7B is constrained by resource limits and attention head count; results are therefore extrapolated using a fitted scaling curve.

Figure 7: **Decoding speed comparison at different output lengths on a CPU-based mobile inference framework.** Results are reported for the 1B-parameter SpikingBrain linear model and the Llama3.2-1B baseline, both evaluated on an Intel Core i5-12600KF CPU. By default, Q4_0 weight quantization is applied to balance throughput and memory efficiency.

## 5.2  Long-context Efficiency

Beyond task performance, we also place particular emphasis on the efficiency of our models in long-sequence inference. Benefiting from the integration of (hybrid) linear attention with the MoE mechanism, both SpikingBrain models exhibit substantial inference speedups. Since the 7B model is more compact and shows stronger efficiency gains, we adapt it to the HuggingFace framework with multi-device sequence-parallel inference (ZeCO + p2p), successfully supporting efficient prefill up to 4M tokens. All measurements reported in Figure 6 and Table 7 are conducted on NVIDIA H100 GPUs and averaged over 10 runs. As shown in Figure 6 and Table 7, at an input length of 1M tokens, the SpikingBrain-7B model achieves a 26.5× speedup in TTFT (Time to First Token) compared with the Qwen2.5 baseline using full attention and all-to-all communication. Due to resource limitations and the number of attention heads, Qwen2.5 is difficult to evaluate at 4M tokens. Based on the fitted scaling curve, we conservatively estimate a speedup of over 100×. Moreover, as sequence length and GPU count increase proportionally, the time overhead of our 7B model remains nearly constant. These results demonstrate the scalability and superiority of our approach in ultra-long-sequence scenarios, providing strong support for the deployment of more efficient large language models in practical applications.

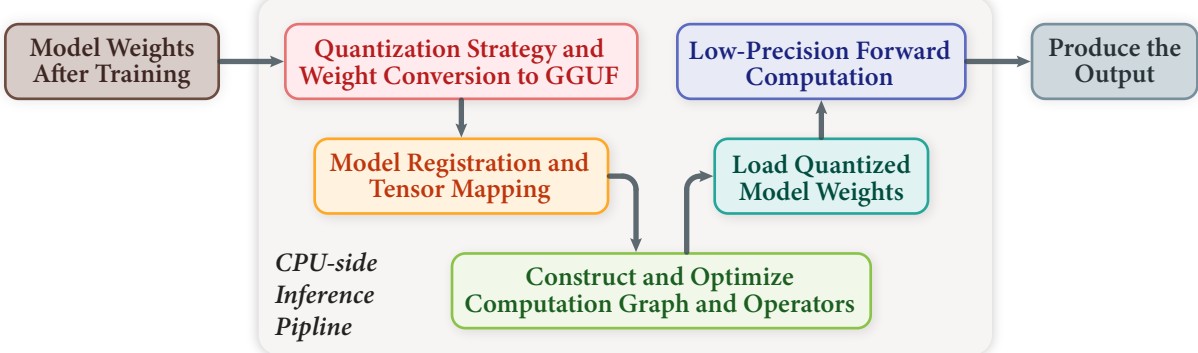

Figure 8: **Overview of the CPU-side inference pipeline.** The workflow includes four main steps: weight conversion and quantization, model registration and tensor mapping, graph and operator optimization, and quantized inference.

In addition, we perform a systematic evaluation of inference speed for both the 7B and 76B models without sequence parallelism enabled. The experiments are conducted on MetaX C550 GPUs, including HuggingFace single-GPU deployment and both single- and multi-GPU deployments using vLLM. As shown in Table 8, SpikingBrain-7B consistently achieves multiplicative speedups over Qwen2.5 at a sequence length of 128k (1.41× on HuggingFace and 2.75× on vLLM), while delivering performance comparable to—or even exceeding—that of the Mistral model, which also employs SWA to achieve linear complexity. For the larger 76B model, inference speed is equally compelling: it not only significantly outperforms Llama2-70B, but even surpasses the MoE baseline Mixtral-8×7B in settings with Expert Parallel (EP) enabled. These results highlight the broad applicability and performance advantages of our approach across model scales and inference frameworks.

For long-sequence training, we further measure Throughput per GPU Second (TGS) on SpikingBrain-7B and the Qwen2.5 baseline under the sequence parallelism framework, using MetaX C550 GPUs. The experiments show a 5.36× throughput advantage for our 7B model at a sequence length of 128k, consistent with the TTFT improvements observed during inference and attributable to its efficient attention design.

### 5.3 CPU-side Inference

In this section, we deploy compressed 1B-parameter models, including the SpikingBrain linear model with SWA and GLA, as well as Llama3.2 (Dubey et al., 2024), on a CPU-based mobile inference framework. The deployment leverages llama.cpp as the inference backend. As illustrated in Figure 8, the workflow consists of four main steps: i) Weight conversion and quantization, ii) Model registration and tensor mapping, iii) Graph and operator optimization, and iv) Quantized inference. See Appendix A.6 for details.

A key innovation of the SpikingBrain-1B linear model lies in its inter-layer hybridization of linear attention and SWA, which avoids full computation over all historical Keys and Values. This design not only reduces KV-cache memory consumption but also significantly lowers inference latency. Compared with conventional Transformer-based models, our computation graph explicitly integrates these mechanisms to further optimize KV-cache management. We evaluate decoding speed at different output lengths using a 1k input. The hardware setup includes an Intel Core i5-12600KF CPU, 64 GB of memory, and Ubuntu 22.04.4 LTS, with the project compiled using CMake 3.28.3. As shown in Figure 7, the SpikingBrain-1B model maintains constant computation and memory overhead during decoding, resulting in stable throughput as the output sequence length increases. In contrast, the Llama3.2-1B baseline shows a sharp decline in decoding speed due to full KV-cache computation. Overall, SpikingBrain-1B achieves speedups of 4.04×, 7.52×, and 15.39× at sequence lengths of 64k, 128k, and 256k, respectively.

This workflow allows the inference engine to seamlessly adapt brain-inspired models for CPU deployment, achieving substantial gains in computational efficiency, memory utilization, and hardware adaptability. It provides a practical solution for running large models efficiently in resource-constrained environments.

## 5.4 Performance on the MetaX Cluster

The performance of the MetaX C550 GPU cluster is evaluated from two perspectives: training efficiency and system stability. For the SpikingBrain-7B model during the training phase, we achieve a TGS (Token Per Second) of 1558 and an MFU (Model FLOPs Utilization) of 23.4% (8-way DP, 4-way PP, PP micro-batch size 2, global batch size 512), reflecting high computational efficiency and effective resource utilization. This result is comparable to the measured MFU of 25.8% on an NVIDIA A800 80GB GPU cluster under the same parallel configuration, despite the use of older software versions on MetaX due to compatibility constraints.

Regarding stability, continuous monitoring of the cluster operation confirms exceptional reliability and robustness. The training session remains uninterrupted for more than two weeks, underscoring the stability and maturity of the MetaX hardware and software ecosystem.

Overall, these results highlight the capability of the MetaX GPU cluster to efficiently and reliably support large-scale long-duration training tasks, enabling stable training of models with tens of billions of parameters. Additional cross-platform experiments comparing end-to-end model performance and operator-level behavior on NVIDIA A100 and MetaX C550 are provided in Appendix A.7.

## 5.5 Analysis of Spiking Scheme

We implement the joint optimization of spiking-based activation encoding and 8-bit fixed-point weight quantization (INT8 Quantization) on our pre-trained models. This combination achieves an effective balance between accuracy and energy efficiency, leveraging event-driven computation paradigm to reduce cost while preserving model performance.

**Scheme Deployment and Accuracy Maintenance** Specifically, we apply spike coding to the activation inputs of all linear projection layers in our models, while the corresponding weights are quantized using symmetric INT8. Quantization parameters are calibrated with a small set of 128 text samples. As shown in Table 3, evaluations on commonsense reasoning, MMLU, and CMMLU benchmarks indicate that the average performance drop under this scheme is limited to approximately 2% for both SpikingBrain-7B and SpikingBrain-76B, confirming its effectiveness in preserving accuracy.

Table 3: **Performance comparison of SpikingBrain before and after applying the spiking scheme.** Evaluations are performed on commonsense reasoning, MMLU, and CMMLU benchmarks. The integration of adaptive-threshold spike coding with 8-bit weight quantization results in negligible performance degradation (approximately 1–3%).

| | Winogrande | Arc-e | Arc-c (norm) | HellaSwag (norm) | PIQA | MMLU | CMMLU | Avg. |
|---|---|---|---|---|---|---|---|---|
| SpikingBrain-7B | 0.6992 | 0.8047 | 0.5566 | 0.6777 | 0.7949 | 0.6751 | 0.6904 | 0.6998 |
| SpikingBrain-7B -W8ASpike | 0.6895 | 0.7861 | 0.5410 | 0.6758 | 0.7979 | 0.6546 | 0.6677 | 0.6875 |
| SpikingBrain -76B | 0.7275 | 0.8125 | 0.5615 | 0.7000 | 0.8125 | 0.7247 | 0.7740 | 0.7304 |
| SpikingBrain -76B-W8ASpike | 0.7148 | 0.7949 | 0.5371 | 0.6863 | 0.8004 | 0.7081 | 0.7512 | 0.7133 |

**Spike Distribution Characteristics** To evaluate the sparsity and efficiency of the proposed spiking scheme, we analyze the statistical firing patterns under the bitwise–ternary coding method. We collect the absolute values of input spike integers across all linear projection layers. The detailed distribution of spike counts is illustrated in Figure 9. Take the SpikingBrain-7B model for example, results show that 94.1% of values fall within [0,7], and only about 1% exceed 16. After expansion into bitwise–ternary spikes, the average number of spikes fired per channel is only 1.13, while 18.4% of channels do not fire any spikes at all. Since 94.1% of values are within 7 (i.e., within the INT3 range), we further analyze the firing rate under a step size of 3. By filling inactive bits within the 3-step window with 0, the overall sparsity reaches about 69.15%.

**Spike Visualization Interface** We provide a unified visualization interface to examine the spiking activity of neurons under different coding strategies. As shown in Figure 10, we compare Binary, Ternary, and Bitwise

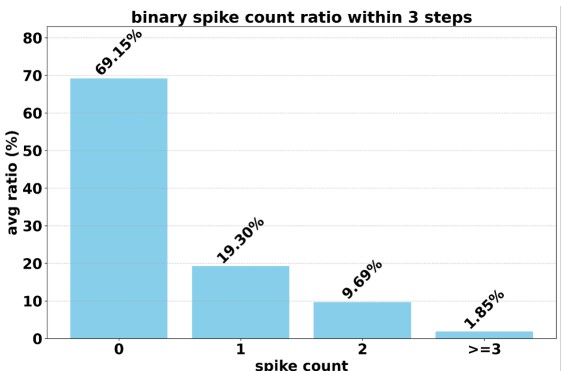 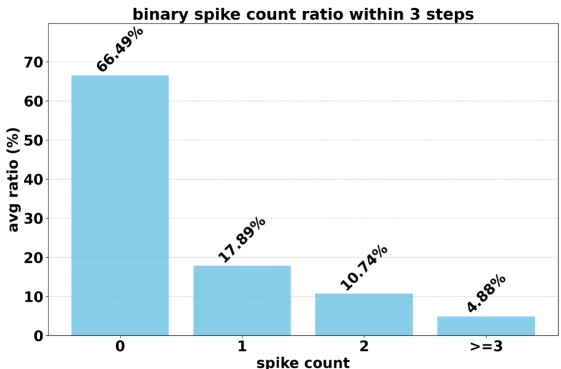

Figure 9: **Spike counts distribution of the bitwise spike coding scheme.** Results are shown for SpikingBrain-7B (left) and SpikingBrain-76B (right).

Spike Coding using two-dimensional time–neuron firing maps. This tool provides an intuitive means for analyzing spiking behaviors across different spike coding strategies:

- **Binary Spike Coding** typically requires longer time steps and higher firing frequencies, resulting in a relatively dense overall spike distribution.

- **Ternary Spike Coding** introduces positive and negative spikes, yielding a significantly sparser representation that reduces both the total spike count and the required time steps.

- **Bitwise Spike Coding** further enhances compression through bit-expansion strategies, substantially reducing time overhead while preserving representational precision.

**Energy Efficiency with Asynchronous Hardware** Statistical analysis shows that during inference, 18.4% of channels remain completely inactive (i.e., emit no spikes). Under the event-driven computing paradigm of asynchronous hardware, all weight-fetching operations for these inactive channels are skipped (including data transfer from off-chip DRAM to on-chip SRAM and from SRAM to compute units), thereby proportionally reducing memory access overhead.

For computation, we estimate and compare the energy consumption of MAC (multiply–accumulate) operations under different paradigms, based on published hardware energy consumption data at 45nm technology (Yao et al., 2024b):

- **Conventional FP16 MAC**: Each operation consumes 1.5 pJ, requiring full floating-point multiplication and addition regardless of whether the activation is effective.

- **INT8 MAC**: Each operation consumes 0.23 pJ, but still depends on synchronous clock-driven execution, incurring energy costs for idle cycles.

- **Proposed scheme (event-driven spiking computing + INT8 weight quantization)**: The MAC energy consumption $E$ is estimated based on "spike-triggered weight additions", i.e.,

$$E = \text{Average Spikes} \times E_{\text{INT8 Add}}. \tag{24}$$

  With an average spike count of 1.13 and INT8 (weight) addition energy consumption of approximately 0.03 pJ per operation, the average MAC energy cost is about 0.034 pJ ($1.13 \times 0.03$ pJ).

The results show that our scheme reduces energy consumption by 97.7% compared with FP16 MACs and by 85.2% compared with INT8 MACs, yielding energy-efficiency improvements of up to $43.48\times$ and $6.76\times$, respectively. This demonstrates the effectiveness of combining spiking-driven computation with quantization to significantly reduce energy overhead, while maintaining precision degradation within a controllable range.

However, these energy-efficiency estimates are based on idealized assumptions regarding asynchronous hardware behavior. Hardware challenges underlying these estimates and corresponding directions toward practical realization are discussed in Appendix C.2.

## 6 Discussion of Negative Results and Practical Challenges

In this section, we discuss experimental configurations and implementation strategies that did not yield the expected improvements, as well as challenges encountered during adaptation to MetaX hardware. Sharing these observations provides practical guidance and improves the transparency and reproducibility of our work.

- **Long-context data construction.** Training on concatenated short texts, as described in Section 3.4, rather than native long-context data, resulted in limited improvements in long-context performance. This suggests that downstream performance at extreme context lengths depends strongly on both the quantity and quality of long-context training data. Incorporating naturally long-context samples or recall-intensive training data is expected to further reduce the performance gap between SpikingBrain and the base Transformer model.

- **MoE upcycling initialization.** Increasing expert diversity through partial random initialization of expert weights did not produce meaningful performance improvements. As a result, we adopt direct copying and rescaling of expert weights during MoE upcycling, which provides stable and consistent performance.

- **Sequence parallelism stability.** During multi-node sequence-parallel (SP) training, we observed reduced GPU utilization and occasional out-of-memory errors or deadlocks. These issues were primarily caused by asynchronous operator accumulation leading to excessive buffer pre-allocation, as well as inter-node load imbalance. We resolved these issues by introducing explicit synchronization after SP communication primitives, which improved both stability and resource utilization.

- **Context length in reasoning SFT.** Using an 8k context length during the reasoning SFT stage resulted in truncation of long chain-of-thought sequences, limiting SpikingBrain's reasoning performance. Increasing the context length (e.g., to 32k) enables more complete reasoning trajectories and improves downstream reasoning capability.

- **Software adaptation on MetaX platform.** Due to software stack incompatibilities on the MetaX platform at the time of development, we adapted the Megatron framework to operate with older versions of PyTorch and Triton. This limited access to newer optimization features and required conservative kernel design choices. For example, reduced Triton block sizes and constrained autotuning configurations were used to ensure stability, but resulted in suboptimal performance compared to fully optimized implementations.

These observations highlight important practical considerations for large-scale deployment and hardware adaptation of efficient and brain-inspired large language models.

## 7 Conclusion

This work provides a comprehensive demonstration of efficient brain-inspired large model training and inference on the MetaX GPU cluster. By integrating several key techniques—including novel (hybrid) linear architectures beyond Transformers, sparse MoE design, adaptive-threshold sparse spiking activation, and lightweight conversion training—we validate and implement a practical development pipeline for spiking-based large models on a non-NVIDIA cluster with hundreds of GPUs.

We release two models as outcomes of this effort: the linear model SpikingBrain-7B and the MoE hybrid-linear model SpikingBrain-76B-A12B. These models offer two main advantages: i) Training efficiency: With linear or near-linear complexity, they substantially accelerate long-sequence training and match the performance of many open-source Transformer models while using less than 2% of the training data via continual pre-training.

ii) Inference performance: During inference, the models exhibit event-driven spiking behavior and maintain constant (SpikingBrain-7B) or layer-wise partially constant (SpikingBrain-76B) memory footprint. This yields order-of-magnitude improvements in long-sequence efficiency and speed (e.g., over $100\times$ TTFT acceleration at 4M tokens).

These results not only demonstrate the feasibility of efficient large-model training on non-NVIDIA platforms, but also outline new directions for the scalable deployment and application of brain-inspired models in future computing systems.

## Broader Impact Statement

Our work introduces SpikingBrain, a framework designed to enhance the efficiency and accessibility of large language models (LLMs) across diverse hardware platforms, including non-NVIDIA GPUs and neuromorphic systems. By significantly reducing the energy consumption associated with LLM training and inference, our methods contribute to the democratization of AI, potentially fostering innovation and research in resource-constrained environments.

However, we recognize that the increased efficiency and broader deployment capabilities of LLMs, facilitated by our framework, also carries dual-use risks. A more widespread accessibility of LLMs could inadvertently amplify existing concerns related to the generation of harmful, biased, or unsafe content. These challenges, while inherent to the scale of LLM technology, are potentially accelerated by tools that simplify their adoption.

We strongly advocate for the responsible development of efficient AI. We encourage users of SpikingBrain to integrate robust safety guardrails and local moderation filters, especially for edge-case deployments. Our ultimate goal is to provide a toolset that balances computational performance with ethical responsibility, ensuring that the transition toward more efficient AI remains aligned with societal well-being.

## Author Contributions

Yuqi Pan and Yuhong Chou designed the model architectures. Siyu Ding proposed the adaptive threshold spiking neuron and conducted spiking-based training. Han Xu and Xuerui Qiu developed three spike coding techniques. Yuqi Pan, Yupeng Feng, and Zehao Liu carried out the framework adaptation on MetaX and model training. Jinghao Zhuang, Yupeng Feng, Zehao Liu, Yuqi Pan, Bohan Sun, and Yuhong Chou performed model evaluation. Anjie Hu and Anlin Deng contributed to code adaptation and refinement. Peng Zhou performed CPU adaptation and model testing. Yuqi Pan, Shurong Wang, Guoqi Li, Man Yao, Jibin Wu, Jian Yang, Guoliang Sun, and Bo Xu contributed to the discussion and analysis of experimental results. The project was supervised by Bo Xu and Guoqi Li.

## Acknowledgments

This work was partially supported by CAS Project for Young Scientists in Basic Research (YSBR-116), National Distinguished Young Scholars (62325603), National Natural Science Foundation of China (62236009, U22A20103), Beijing Science and Technology Plan (Z241100004224011), and Shanghai Neuhelium Neuromorphic Intelligence Technology Co.Ltd.

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

# A  Experiments

## A.1  Benchmarks

In selecting evaluation metrics, we place greater emphasis on pretraining-oriented general-purpose benchmarks: MMLU (Hendrycks et al., 2020), CMMLU (Li et al., 2023a), C-Eval (Huang et al., 2023), ARC-C (Clark et al., 2018), and HellaSwag (Zellers et al., 2019), as these better indicate whether our models—trained with fewer than 200B tokens during efficient conversion—can inherit the knowledge and modeling capacity of the base model. Moreover, these benchmarks reveal whether the converted models preserve strong generalization while demonstrating significant long-sequence efficiency advantages, without being confounded by data quality factors such as alignment performance after SFT.

For SpikingBrain chat models after three-stage SFT, we evaluate general knowledge, long-sequence modeling, and instruction following, additionally including NQ (Kwiatkowski et al., 2019), TrQ (Joshi et al., 2017), QuALITY (Pang et al., 2022), and IFEval (Zhou et al., 2023). To assess real-world long-context modeling and understanding capability, we also evaluate six popular tasks on LongBench (Bai et al., 2024).

## A.2  Models

We compare SpikingBrain models against open-source baselines of comparable scale spanning Hybrid, Transformer, and MoE architectures:

- **Linear-complexity models**: Falcon-Mamba-7B (Zuo et al., 2024), Mistral-7B (team, 2023).

- **Hybrid linear-complexity models**: Zamba-v1-7B (Glorioso et al., 2024), Jamba-52B-A12B (Lieber et al., 2024).

- **Quadratic-complexity models**: Llama3.1-8B (Dubey et al., 2024), Qwen2.5-7B (Yang et al., 2024b), Mixtral-47B-A13B (Jiang et al., 2024), LLama2-70b (Touvron et al., 2023), Gemma2-27B (Team et al., 2024).

## A.3  Evaluation of Instruct Models

We apply three-stage SFT to the converted models to add instruction following. As shown in Table 4, the aligned models match similarly sized open-source chat baselines across general knowledge, long-sequence modeling, and instruction following, while avoiding overfitting and preserving pretrained capabilities—indicating that (hybrid) linear architectures remain stable and scalable under alignment.

Table 4: **Performance Evaluation of SpikingBrain Instruct Models.** All models are tested with the vLLM framework and evaluated using a generation-based method. For QuALITY and IFEval, we report results from the non-CoT model (after SFT stage 2) to avoid chain-of-thought interference.

| | SpikingBrain-7B | SpikingBrain-76B | Llama3 | Qwen2.5 | Mixtral |
|---|---|---|---|---|---|
| Params | 7B | 12B/76B | 8B | 7B | 13B/47B |
| Complexity Type | Linear | Hybrid | Quadratic | Quadratic | Quadratic |
| **Benchmarks** | | | | | |
| MMLU | 65.57 | 73.71 | 68.69 | **75.17** | 71.03 |
| CMMLU | 68.76 | 77.41 | 55.17 | **79.14** | 51.03 |
| HellaSwag | 68.95 | **86.63** | 76.80 | 85.39 | 75.63 |
| Ceval | 69.07 | 76.32 | 55.01 | **77.93** | 50.88 |
| NQ | 21.47 | 21.55 | **30.97** | 17.67 | 28.48 |
| TrQ | 57.03 | 55.13 | 65.78 | 55.72 | **71.00** |
| QuALITY | 60.12 | 69.56 | 66.25 | **73.63** | 51.34 |
| IFEval | 42.70 | 49.72 | 73.01 | **73.20** | 48.06 |

### A.4 Evaluation on LongBench

To assess real-world long-context understanding, we evaluate SpikingBrain-76B (after stage-2 SFT) on six LongBench tasks (Bai et al., 2024), including HotpotQA, NarrativeQA, TriviaQA, TREC, MultiNews and Qasper. Given the baseline's maximum positional embeddings, we truncate prompts to 32k. As shown in Table 5, our model is competitive with other hybrid and Transformer baselines while using only four global softmax attention layers.

Table 5: **Performance Evaluation of SpikingBrain-76B on six LongBench tasks (32k).**

|  | **SpikingBrain-76B** | Jamba | Qwen2.5 | Mixtral |
|---|---|---|---|---|
| Params | 12B/76B | 12B/52B | 7B | 13B/47B |
| Complexity Type | Hybrid | Hybrid | Quadratic | Quadratic |
| **Benchmarks** | | | | |
| HotpotQA | 50.00 | 11.38 | **57.62** | 36.42 |
| NarrativeQA | 23.92 | 3.73 | **27.96** | 7.44 |
| TriviaQA | 90.84 | **92.63** | 85.92 | 85.97 |
| TREC | 67.00 | 77.00 | **68.50** | 19.25 |
| MultiNews | 25.89 | 8.44 | 23.79 | **26.52** |
| Qasper | 39.59 | 10.67 | **45.13** | 28.37 |

To strengthen the comparison, we extend the Qwen2.5 baseline to 128k context length using YaRN and evaluate the same six LongBench tasks with prompts truncated to 128k tokens. As shown in Table 6, the results demonstrate that SpikingBrain-76B remains competitive at longer context lengths, even with a relatively high hybrid ratio of 1:6.

Table 6: **Performance Evaluation of SpikingBrain-76B on six LongBench tasks (128k).**

|  | **SpikingBrain-76B** | Qwen2.5 |
|---|---|---|
| Params | 12B/76B | 7B |
| Complexity Type | Hybrid | Quadratic |
| **Benchmarks** | | |
| HotpotQA | 48.42 | **57.96** |
| NarrativeQA | 23.35 | **27.10** |
| TriviaQA | **91.24** | 88.25 |
| TREC | **72.00** | 62.50 |
| MultiNews | **26.51** | 23.51 |
| Qasper | 38.85 | **44.43** |

**Discussion of Recall.** It is important to note that LongBench primarily evaluates long-context modeling and understanding capabilities, and does not directly measure long-context recall performance. Linear and subquadratic (hybrid) attention models exhibit known limitations in recall due to their reduced effective state size compared to quadratic Transformers (Waleffe et al., 2024; Arora et al., 2024a). We empirically validate this recall gap in Appendix B.1.

This limitation can be mitigated through improved training data quality and scale. Prior industrial studies (Qwen Team, 2025; Team et al., 2025) have shown that hybrid attention architectures can achieve retrieval performance comparable to Transformers when trained with sufficient data and infrastructure. These findings suggest that recall limitations are not intrinsic barriers, but can be reduced through appropriate training and system design.

## A.5 Long-context Efficiency

Table 7: **Inference speed comparison of SpikingBrain-7B under sequence parallelism (SP).** The metric is TTFT (ms), defined as the latency to complete prefill and generate the first token for a given prompt length. In SP configuration, our 7B model uses ZeCO for the linear attention module and P2P communication for the SWA module, while the Qwen2.5 baseline employs A2A communication. All measurements are conducted on NVIDIA H100 GPUs and averaged over 10 runs. "−" indicates infeasible measurement due to resource constraints.

| Sequence Length | 256k | 512k | 1M | 2M | 4M |
| GPU Count | 8 | 16 | 32 | 64 | 128 |
| --- | --- | --- | --- | --- | --- |
| **SpikingBrain-7B** | **1015** | **1037** | **1054** | **1070** | **1073** |
| Qwen2.5 | 7419 | 14398 | 27929 | − | − |

Table 8: **Inference latency (ms) comparison across models, frameworks, and sequence lengths.** Tests are conducted on HuggingFace single-GPU deployment with MetaX C550 GPUs and on vLLM with both single- and multi-GPU settings. Latency is measured as the time to process the input sequence and generate 128 output tokens.

| | SpikingBrain -7B | Mistral -7B-v0.1 | Qwen2.5 -7B | SpikingBrain -7B | Mistral -7B-v0.1 | Qwen2.5 -7B | SpikingBrain -76B | Mixtral -8x7B-v0.1 | Llama -2-70b |
| --- | --- | --- | --- | --- | --- | --- | --- | --- | --- |
| GPUs | 1 | | | 1 | | | 4 | | |
| Frameworks | HuggingFace | | | vllm | | | vllm | | |
| Expert Parallel | Non-MoE | | | | | | Enable / Disable | | Non-MoE |
| 32k | 8452 | 10202 | 7969 | 4369 | 4570 | 6421 | 9451 / 5305 | 10477 / 3209 | 15881 |
| 64k | 11685 | 16467 | 16257 | 7399 | 7084 | 15093 | 14077 / 8142 | 19062 / 4840 | 32953 |
| 128k | 15974 | 28382 | 38487 | 12048 | 11867 | 45141 | 27106 / 14109 | 34983 / 8046 | 86120 |

## A.6 CPU-side Inference

We demonstrate the deployment of compressed 1B-scale models, including the SpikingBrain linear model with SWA and GLA, as well as Llama3.2 (Dubey et al., 2024), on a CPU-based mobile inference framework. The deployment leverages llama.cpp as the inference backend. As illustrated in Figure 8, the workflow consists of four main steps:

- **Weight conversion and quantization**. The trained weights are first converted into the GGUF format with an appropriate quantization strategy. Unlike generic training formats, GGUF emphasizes compact encoding and cross-platform consistency, yielding more predictable load latency and memory footprint during CPU inference. Floating-point weights are compressed into low-bit integers (e.g., Q4, Q5). Combined with efficient decoding and optimized matrix-multiplication kernels, this greatly improves throughput while reducing memory usage.

- **Model registration and tensor mapping**. A dedicated computation graph is constructed by registering architecture components and mapping tensors within the inference engine. Each tensor in the weight file is directly linked to its corresponding graph structure, with associated metadata such as shape and precision format (e.g., float32, float64).

- **Graph and operator optimization**. Following the forward propagation flow, we implement key operators including attention, rotary position embedding (RoPE (Su et al., 2021)), and residual connections. Two major optimizations are applied: i) Operator fusion: combining matrix multiplication, bias addition, and nonlinear activation into a single kernel to reduce intermediate storage and memory access. ii) Graph-level optimization: decomposing and reorganizing operations such as RMSNorm, Softmax, and RoPE to enable fusion with matrix multiplication or low-level kernels, minimizing redundant computation.

- **Quantized inference**. Finally, the quantized weights are loaded, and forward computation is performed in low precision. By default, Q4_0 quantization is applied to the weights to balance computational throughput with memory usage. In addition, we use compact single-step integer activations rather than expanded spike trains, ensuring compatibility with current CPU hardware.

A key innovation of the SpikingBrain-1B linear model lies in its inter-layer hybridization of linear attention and SWA, which avoids full computation over all historical Keys and Values. This design not only reduces KV-cache memory consumption but also significantly lowers inference latency. Compared with conventional Transformer-based models, our computation graph explicitly integrates these mechanisms to further optimize KV-cache management. We evaluate decoding speed at different output lengths using a 1k input. The hardware setup includes an Intel Core i5-12600KF CPU, 64 GB of memory, and Ubuntu 22.04.4 LTS, with the project compiled using CMake 3.28.3. As shown in Figure 7, the SpikingBrain-1B model maintains constant computation and memory overhead during decoding, resulting in stable throughput as the output sequence length increases. In contrast, the Llama3.2-1B baseline shows a sharp decline in decoding speed due to full KV-cache computation. Overall, SpikingBrain-1B achieves speedups of 4.04×, 7.52×, and 15.39× at sequence lengths of 64k, 128k, and 256k, respectively.

This workflow allows the inference engine to seamlessly adapt brain-inspired models for CPU deployment, achieving substantial gains in computational efficiency, memory utilization, and hardware adaptability. It provides a practical solution for running large models efficiently in resource-constrained environments.

### A.7 MetaX Performance Metrics Relative to NVIDIA

Beyond the MFU comparison with NVIDIA A800, we conduct additional experiments on both NVIDIA A100-SXM4-80GB and MetaX C550 (64GB HBM), focusing on end-to-end model performance and operator-level behavior. These experiments aim to demonstrate the cross-platform adaptability of SpikingBrain and enable direct performance comparison under controlled conditions.

**Overall model speed comparison.** We benchmark forward-only execution of SpikingBrain-7B with batch size (bs) = 1 and sequence lengths ranging from 1k to 16k. For each configuration, throughput (tokens/s) is computed as bs × seqlen/wall-clock time. Identical runtime settings and measurement protocols are used on both platforms, including warm-up runs followed by averaged timed measurements.

As shown in Table 9, MetaX achieves approximately 0.77×–0.93× the throughput of NVIDIA A100 in this forward benchmark. Notably, the performance gap narrows as sequence length increases, indicating improved hardware utilization on MetaX in long-context regimes.

Table 9: **Forward throughput comparison of SpikingBrain-7B on NVIDIA A100 and MetaX C550.** Throughput is measured in tokens/s with batch size = 1.

| Sequence Length | NVIDIA (tokens/s) ↑ | MetaX (tokens/s) ↑ | Ratio |
|---|---|---|---|
| 1024 | 11140.93 | 8265.79 | 0.769 |
| 2048 | 12316.52 | 10440.11 | 0.848 |
| 4096 | 12524.46 | 10897.70 | 0.870 |
| 8192 | 12621.84 | 11392.21 | 0.903 |
| 16384 | 12768.80 | 11847.01 | 0.928 |

**Operator-level comparison.** To further analyze hardware efficiency, we profile an inter-layer hybrid block and measure per-operator latency using GPU event timing. To ensure fair shape comparison, we fix the total token count (bs × seqlen = 65536) while varying batch size and sequence length. All latency values are reported in milliseconds.

As shown in Table 10, the end-to-end hybrid block latency remains within ±2% between MetaX C550 and NVIDIA A100 across all tested configurations. At operator granularity, MetaX is consistently faster on FFN and normalization layers, largely comparable on SWA, while linear attention (LA) remains the primary source of latency overhead on MetaX. However, this gap decreases as sequence length increases.

Table 10: **Operator-level latency comparison of SpikingBrain inter-layer hybrid blocks on NVIDIA A100 and MetaX C550.** All values are in milliseconds.

| bs | seqlen | NVIDIA (ms) ↓ | MetaX (ms) ↓ | Δ total | Δ FFN | Δ LA | Δ SWA | Δ Norm |
|----|--------|---------------|--------------|---------|-------|------|-------|--------|
| 16 | 4096   | 7051 | 7188 | +1.9% | -6.2% | +54.6% | -4.2% | -2.9% |
| 8  | 8192   | 7220 | 7264 | +0.6% | -6.0% | +45.8% | -6.0% | -2.9% |
| 4  | 16384  | 7284 | 7236 | -0.7% | -5.8% | +33.7% | -5.1% | -2.7% |
| 2  | 32768  | 7303 | 7284 | -0.3% | -5.7% | +33.6% | -3.2% | -2.7% |
| 1  | 65536  | 7291 | 7258 | -0.5% | -6.1% | +23.9% | +4.2% | -2.9% |

These results support several important observations. First, SpikingBrain demonstrates strong cross-platform portability, achieving comparable end-to-end performance on both NVIDIA A100 and MetaX C550 using the same implementation. The hybrid block latency remains within $\pm 2\%$ across a wide range of tensor shapes. Second, the throughput gap between MetaX and NVIDIA decreases as sequence length increases, with MetaX reaching $0.928\times$ of NVIDIA throughput at 16k sequence length, compared to $0.769\times$ at 1k, indicating favorable scaling in long-context settings. Finally, operator-level analysis reveals that MetaX outperforms NVIDIA on FFN and normalization layers, while the remaining performance gap is primarily attributable to linear attention. This suggests that further optimization of linear attention kernels represents the most direct path to closing the remaining performance gap.

## A.8 Effect of Adaptive Threshold Hyperparameter

The adaptive threshold hyperparameter $k$ controls the scaling of the firing threshold and thereby determines the spike count distribution. Specifically, the adaptive threshold is defined as

$$V_{\text{th}}(\mathbf{x}) = \frac{1}{k}\,\text{mean}(\text{abs}(\mathbf{x})), \tag{25}$$

such that larger values of $k$ correspond to lower thresholds, resulting in increased spike activity (i.e., reduced sparsity). While lower sparsity generally improves accuracy by preserving more activation information, it also reduces the potential efficiency benefits of spike-based computation. This reflects an inherent accuracy–sparsity trade-off in spike coding.

We empirically evaluate this trade-off using the SpikingBrain-7B model across multiple benchmark tasks. As shown in Table 11, increasing $k$ consistently improves overall accuracy while decreasing sparsity, confirming the expected relationship between threshold scaling, spike activity, and model performance. Based on this trade-off, we select $k = 3$ as the default setting, which provides a favorable balance between accuracy and sparsity.

Table 11: **Effect of adaptive threshold hyperparameter** $k$ **on accuracy and sparsity in SpikingBrain-7B.** Larger $k$ values reduce sparsity and improve accuracy, demonstrating the accuracy–sparsity trade-off.

| Configuration | WG | ARC-E | ARC-C | HellaSwag | PIQA | MMLU | CMMLU | Avg. | Sparsity | Values ≤ 7 |
|---------------|------|-------|-------|-----------|------|------|-------|------|----------|-----------|
| BF16 | 0.6992 | 0.8047 | 0.5566 | 0.6777 | 0.7949 | 0.6751 | 0.6904 | 0.6998 | – | – |
| W8ASpike ($k=2$) | 0.6835 | 0.7887 | 0.5213 | 0.6931 | 0.7927 | 0.6012 | 0.6339 | 0.6735 | 0.730 | 0.989 |
| W8ASpike ($k=2.5$) | 0.6780 | 0.7934 | 0.5316 | 0.7021 | 0.7894 | 0.6141 | 0.6560 | 0.6807 | 0.709 | 0.976 |
| **W8ASpike ($k=3$)** | 0.6895 | 0.7861 | 0.5410 | 0.6758 | 0.7979 | 0.6546 | 0.6677 | 0.6875 | 0.692 | 0.951 |
| W8ASpike ($k=3.5$) | 0.6851 | 0.8035 | 0.5427 | 0.7012 | 0.7933 | 0.6305 | 0.6723 | 0.6898 | 0.677 | 0.916 |
| W8ASpike ($k=4$) | 0.6946 | 0.8056 | 0.5452 | 0.7046 | 0.7933 | 0.6331 | 0.6817 | 0.6940 | 0.664 | 0.877 |

## A.9 Spike Visualization Interface

We provide a unified visualization interface to examine the spiking activity of neurons under different coding strategies. As shown in Figure 10, we compare Binary, Ternary, and Bitwise Spike Coding using two-dimensional time–neuron firing maps. This tool provides an intuitive means for analyzing spiking behaviors across different spike coding strategies.

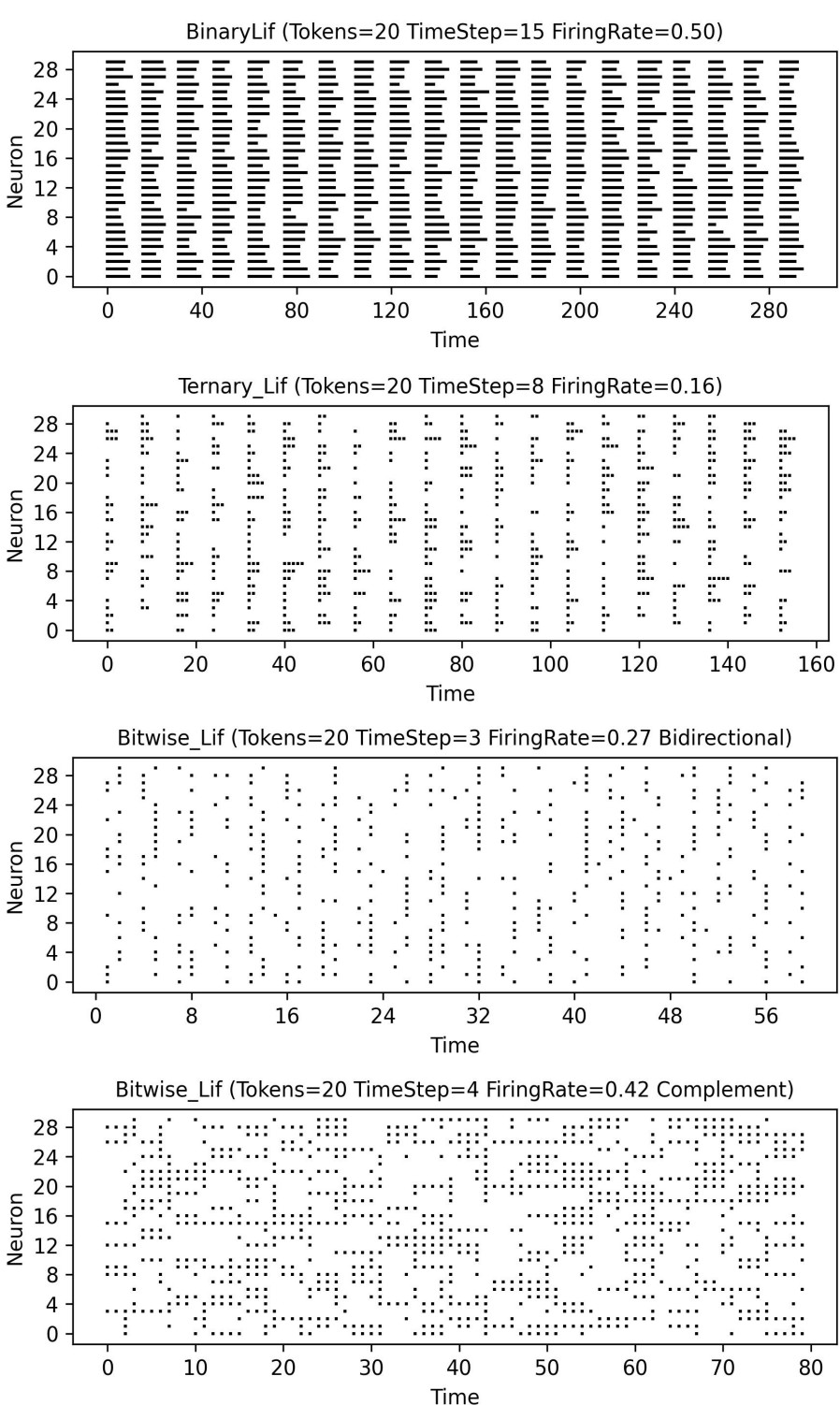

Figure 10: **Time–neuron firing maps under different spike coding schemes.** The figure shows the spike firing distributions for the same input under different coding strategies, including Binary, Ternary, and two variants of Bitwise spike coding. The horizontal axis represents time (Time), defined as token timesteps × expanded timesteps; the vertical axis represents neuron index (Neuron). Black dots indicate spike events of the corresponding neuron at that time.

# B    Ablation Studies

To enable controlled comparisons, we conduct strict data-matched conversion ablation experiments using continual pre-training (CPT) from Qwen3-1.7B with 5B tokens. The training configuration uses sequence length 8k, batch size 32, and learning rate $3 \times 10^{-5}$.

## B.1    Attention Comparison under Identical Conversion Data Budgets

We compare the inter-layer hybrid attention paradigm used in SpikingBrain-7B (LA:SWA = 1:1), pure linear attention (LA, similar to Falcon-Mamba), and pure sliding-window attention (SWA, similar to Mistral) under identical conversion settings. For SWA training, we use a window size of 4k and report the equivalent window size for all models. For fair comparison, we additionally evaluate a pure SWA variant with the window size fixed to 128.

Table 12: **Attention comparison under identical conversion data budgets.**

| Model | Equivalent Window Size | HellaSwag | MMLU | SNIAH1-4k | SNIAH1-8k |
|---|---|---|---|---|---|
| FA Baseline | $\infty$ | 60.21 | 61.78 | 100.0 | 98.41 |
| Pure SWA (4k) | 4096 | 60.12 | 61.74 | 100.0 | 47.60 |
| Pure SWA (128) | 128 | – | – | 1.87 | 0.00 |
| Pure LA | 64 | 54.72 | 39.87 | 36.16 | 39.40 |
| Hybrid | 2080 | 57.88 | 56.08 | 96.83 | 44.44 |

The results in Table 12 reveal several important trends:

- **General modeling performance is strongly influenced by the degree of architectural deviation from the original Transformer.** Pure SWA achieves performance closest to the full-attention baseline on HellaSwag and MMLU, indicating that smaller architectural changes facilitate faster recovery of language modeling capability under limited conversion data.

- **Recall performance is primarily governed by effective state size.** While SWA-4k achieves optimal recall within its attention window, pure linear attention demonstrates substantially stronger recall than SWA-128 despite having comparable state size, reflecting its ability to accumulate information beyond fixed window constraints.

- **Hybrid attention provides the best overall trade-off between performance, recall capability, and efficiency.** By combining SWA and linear attention, the hybrid design achieves strong recall performance while maintaining favorable computational efficiency and memory usage, making it the most effective architecture choice under constrained conversion settings.

## B.2    Isolated Impact of Hybrid Attention, MoE, and Spike Coding

SpikingBrain consists of three core components: hybrid efficient attention (H), MoE modules (M), and spike encoding (S). Together, these components enable competitive performance, improved long-context efficiency, hardware adaptability, and potential energy-efficiency benefits.

For comprehensive ablation analysis, we evaluate all seven possible combinations of these components: H, M, S, H+M, H+S, M+S, and H+M+S. The components are defined as follows:

- **Hybrid efficient attention (H):** inter-layer hybridization of linear attention and sliding-window attention with a 1:3 ratio.

- **MoE modules (M):** sparse mixture-of-experts with 4 routed experts (top-1 routing) and one shared expert.

- **Spike encoding (S):** adaptive-threshold spike encoding applied to activations.

- **H+M+S:** the full SpikingBrain model.

We evaluate both model performance and long-context efficiency for all configurations, together with the Qwen3-1.7B baseline.

Table 13: **Component ablation study of SpikingBrain.** Note: Values in parentheses indicate configurations where spike encoding uses integer activations on GPUs rather than neuromorphic hardware. In this setting, TTFT is identical to the corresponding non-spike configuration.

| Model | HellaSwag ↑ | MMLU ↑ | TTFT-64k (ms) ↓ | TTFT-128k (ms) ↓ |
|-------|-------------|--------|-----------------|------------------|
| Baseline | 60.21 | 61.78 | 3805.6 | 12553.3 |
| H | 57.88 | 56.08 | 2239.7 | 4729.4 |
| M | 60.25 | 61.88 | 4880.7 | 14230.1 |
| S | 59.63 | 61.17 | (3805.6) | (12553.3) |
| H+M | 57.93 | 56.05 | 2558.2 | 5055.2 |
| H+S | 57.37 | 54.87 | (2239.7) | (4729.4) |
| M+S | 59.66 | 59.22 | (4880.7) | (14230.1) |
| H+M+S | 57.77 | 55.02 | (2558.2) | (5055.2) |

The results in Table 13 highlight the distinct contributions of each component:

- **Hybrid efficient attention** is the dominant contributor to long-context efficiency gains, reducing TTFT by more than 2×, albeit with some performance degradation.

- **MoE modules** improve model accuracy with relatively modest efficiency impact. In this ablation setting, the performance gains from MoE are moderate due to the limited conversion data, but larger gains are expected with increased training scale.

- **Spike encoding** consistently incurs only limited performance degradation across configurations, while offering potential energy-efficiency benefits under appropriate hardware assumptions.

Overall, these results suggest a modular adoption strategy: Hybrid attention is the primary choice when long-context efficiency is critical; MoE can be incorporated to further improve model performance; and spike encoding is particularly suitable for deployments targeting event-driven or asynchronous hardware.

## C  Hardware Implementation of Spike Coding

As described in Section 3.3, the proposed spike coding scheme converts continuous-valued activations into two equivalent representations: integer spike counts and expanded sparse spike trains. The integer formulation enables efficient execution on conventional GPU hardware, while the spike-train formulation naturally supports event-driven execution paradigms.

When used purely for numerical simulation, this representation requires no specialized hardware support and serves primarily for algorithmic validation. However, achieving practical performance and energy-efficiency benefits requires careful hardware-aware implementation. In this section, we discuss implementation considerations for both GPU platforms and asynchronous neuromorphic hardware.

### C.1  GPU-based Implementation

On GPUs, acceleration is achieved by leveraging INT-type quantized kernels. In SpikingBrain, this is enabled by INT8 quantization of weights (Section 5.5), allowing spike-count-based integer operations to be executed efficiently using existing GPU kernel optimizations. This formulation preserves compatibility with conventional deep learning software stacks while enabling efficient computation through integer arithmetic and reduced memory bandwidth.

### C.2 Asynchronous Neuromorphic Hardware Implementation

The energy-efficiency estimates presented in Section 5.5 are based on idealized assumptions regarding asynchronous hardware behavior. In practice, the realized efficiency depends on hardware architecture, memory organization, and control mechanisms. Achieving effective energy savings therefore requires careful algorithm–hardware co-design. Below, we discuss key hardware challenges and possible implementation strategies.

**Hardware challenges underlying the theoretical estimates:**

- **Memory access granularity.** The analysis assumes channel-level skipping. In practice, DRAM/SRAM operate on block granularity; if active and inactive channel weights are co-located, memory fetches cannot be avoided, reducing effective savings.

- **Asynchronous control overhead.** The estimation excludes the energy and area cost of asynchronous control logic (e.g., handshake protocols, spike detection, event routing), which introduces additional signal switching absent in synchronous designs.

- **Dynamic spike sparsity.** The average spike count 1.13 is dataset- and layer-dependent. Variations in sparsity across inputs and layers may reduce efficiency relative to the estimated average. Hardware designed for fixed average sparsity will lose energy efficiency in low sparsity scenarios.

**Directions toward practical realization:**

- **Algorithm–hardware co-design for memory access.** Conduct offline profiling to identify sparsity pattern of SpikingBrain. On the software side, algorithm-level weight rearrangement and lightweight compression (e.g., run-length encoding) could be performed; on the hardware side, a channel-aware memory controller could be designed and channel-level memory partitioning could be implemented to isolate weights of active and inactive channels into separate memory blocks, enabling effective skipping of inactive channels.

- **Low-overhead asynchronous control.** Adopt lightweight asynchronous protocols to reduce control overhead. For instance, click controllers and two-phase handshake protocols can minimize unnecessary signal switching. Additionally, an adaptive scheduler could be implemented to monitor spike sparsity in real time. The scheduler dynamically decides whether to merge low-activity channels or expand computing resources, further optimizing energy consumption.

- **End-to-end hardware modeling and optimization.** In hardware development, the theoretical estimation model is gradually revised by incorporating previously omitted overheads derived from hardware simulation into the model. The revised model can be used to guide hardware design and algorithm optimization, ensuring that the balance of power, performance and area is achieved, energy efficiency claims are consistent with practical implementation results, and a complete estimation of end-to-end energy consumption is realized.

