# OpenReview forum: "SpikingBrain: Spiking Brain-inspired Large Models"
_TMLR — Accepted by TMLR_

### Review · Reviewer_WkSJ · 2025-12-16

**Summary Of Contributions:**

The paper proposes SpikingBrain, a novel brain-inspired LLM architecture and its training recipe on MetaX GPUs. The model architecture involves hybrid-linear attention with adaptive spiking neurons for efficient long-context training and inference. The new architecture is compatible with the standard transformer architecture and the architectures of many existing pre-trained LLMs: one can convert an existing pre-trained open-weight LLM into a SpikingBrain model with very little amount of fine-tuning. The authors also released their customized training and inference frameworks for LLMs on MetaX GPUs. The resulting SpikingBrain-7B model significantly improves on long-context efficiency over existing open-weight pre-trained LLMs while retaining most of the performance.

Key strengths:
- Proposes a novel brain-inspired LLM architecture that is compatible with existing mainstream pre-trained LLMs and therefore does not require pre-training from scratch which is computationally prohibitive for most academic researchers.
- The proposed architecture enables efficient long-context training and inference. This is supported with strong empirical evidence.

Key weaknesses:
- Some descriptions about the training and evaluation recipe are somewhat vague and non-concrete, such as the bullet points on pages 8-9.
- Some headline claims rely on indirect or hypothetical evidence. For example, the 100x TTFT speedup for 4M-token is estimated with extrapolated results on the baseline Qwen model. The energy efficiency claims on asynchronous hardware are also based on hypothetical and highly idealized estimations.

**Audience:**

Yes

**Audience Explanation:**

The findings of the paper is highly relevant to the machine learning community, especially researchers working with LLMs. The paper proposes a new type of LLM architecture with efficient conversion from existing open-weight pre-trained models, and introduces effective fine-tuning recipes of LLMs (with MoE and spike coding) on non-NVIDIA hardware. These contributions provides new insights for researchers to explore novel LLM architecture designs, and for practitioners to train LLMs with MoE more effectively.

**Broader Impact Concerns:**

The methods introduced by the paper enables more efficient training and inference of LLMs on non-NVIDIA GPUs and potentially new types of hardware. While it broadens the access of LLMs, it can also potentially increase risk by supporting more widespread deployment of LLMs without safety controls. Although this risk is not unique to this work, I would encourage the authors to add a Broader Impact Statement.

**Claims And Evidence:**

Yes

**Claims Explanation:**

The core results and claims of the paper are mostly well-supported by convincing evidence.
- The paper reports benchmark results for SpikingBrain models against a range of baselines, and describes the multi-stage CPT+SFT pipeline used to adapt from a Transformer checkpoint with substantially fewer tokens than training-from-scratch regimes.
- The reported spike distribution characteristics supports the claim that the encoding yields meaningful sparsity.
- Results in table 3 supports the claim that applying the spiking scheme retains most of the performance on benchmarking tasks.

**Requested Changes:**

Critical changes:
- Provide more concrete details about the training and evaluation recipe. This is most applicable for the bullet points in pages 8-9, but also applicable in some other cases (like the bullet points on pages 15-16). For example, for "perform[ing] long-context extension during conversion," how is it exactly done? When "restrict[ing] the training length and resource usage of the original quadratic attention model," what is the exact amount of  compute/memory that you restrict the model to?
- For Table 1 and Table 2, what does the bolded numbers refer to? I believe conventionally they refer to the best result in each row/column, but it does not seem to be the case here, and can be a bit misleading to the readers.

Non-critical changes that strengthen the work:
- Adding more ablations experiments that isolate contributions. E.g. performance of model converted without spike coding vs. with spike coding; effect of adaptive threshold hyperparameter k on sparsity/accuracy.
- More details on MetaX training stability/utilization. How is the MFU (Model FLOPs Utilization) defined? What are the failure/restart rates across all the training runs?

---

### Review · Reviewer_gn98 · 2025-12-24

**Summary Of Contributions:**

This paper presents SpikingBrain, a suite of large language models that incorporate hybrid efficient attention mechanisms (linear attention combined with sliding-window attention), MoE upcycling, and a novel adaptive-threshold spiking technique to achieve activation sparsity. A core contribution is the demonstration that both a 7B model and a 76B-parameter MoE model can be trained and deployed on MetaX, a non-NVIDIA GPU platform, using a conversion-based training pipeline that repurposes existing Transformer checkpoints. The authors report strong long-context efficiency, most notably >100× TTFT speedups at 4M-token contexts and near-constant memory inference, along with preservation of roughly 90% of the base model's capability. The work is technically ambitious with strong systems engineering depth, covering architecture design, distributed training optimizations, operator adaptations, and multi-platform deployment. However, several scientific claims require further empirical support, and some framing, particularly around biological analogies, should be moderated to better reflect the engineering nature of the contributions.

**Audience:**

Yes

**Audience Explanation:**

Yes.

At least some individuals in TMLR’s audience would likely be very interested in the findings of this paper, for systems and engineering contributions, long-context LLM efficiency, spiking / neuromorphic inspiration

**Broader Impact Concerns:**

No Concerns

**Claims And Evidence:**

Yes

**Claims Explanation:**

Partially.

The submission provides convincing evidence for its systems-level and efficiency-related claims, but the support for several scientific and architectural claims is incomplete or insufficiently validated.

**Requested Changes:**

1. Provide Comprehensive Ablation Studies

The paper combines three distinct techniques (hybrid attention, MoE, and spiking) but does not disentangle their individual contributions. Please provide ablation experiments that isolate the impact of:

• Hybrid linear/SWA attention alone

• MoE upcycling alone

• Spiking activation alone

• Pairwise combinations of the above

For each configuration, report both performance metrics (MMLU, CMMLU, etc.) and efficiency metrics (TTFT, memory usage, throughput). This will allow readers to understand which component drives which benefit and guide practitioners on which techniques to adopt for their specific use cases.

2. Clarification on “Training From Scratch”

The conversion framework claims ~2% of data is sufficient, but only because the model starts from a full Transformer checkpoint. Could the proposed hybrid architecture converge if trained from scratch with this budget?

3. Moderate Biological Claims and Reframe as Engineering Contributions

The extensive emphasis on "brain-inspired" mechanisms and biological plausibility is not well-supported by the technical content. Claims such as "linear attention models dendritic dynamics" or "adaptive threshold preserves biological plausibility" are loose analogies rather than mechanistically accurate descriptions. The adaptive threshold V_th = mean(abs(x))/k is a practical statistical heuristic, not derived from biophysical principles. The neuroscience metaphors should be moderated or explicitly reframed as inspirational analogies, not mechanistic claims. There is no neuroscientific validation, and overstated claims may detract from the genuine engineering contributions.

4. Provide MetaX performance metrics relative to NVIDIA

Since MetaX operators are a key contribution, it would be helpful to include Direct performance comparison: MetaX C550 vs NVIDIA A100/H100 on identical model configurations, operator-level breakdowns showing where MetaX excels or lags. This will help readers assess the generalizability beyond MetaX-specific environments.

5. Clarify near-term utility of spiking on today’s hardware

Given that spiking is simulated and introduces accuracy tradeoffs: What is the immediate value proposition for practitioners using current GPU stacks?
A clearer statement of practical implications would strengthen the contribution.

6. Long-context evaluation at 4M tokens

Beyond TTFT throughput, please include qualitative or accuracy-based evaluations at million-token scales (e.g., Needle-in-a-Haystack, long-range QA) to confirm that the speedups do not come at the cost of severe retrieval degradation.

Minor Recommendation

1. Consider code and model release for reproducibility
2. Consider adding a section on things that did not work.

Verdict: Major Revision

---

### Review · Reviewer_WxpP · 2026-01-31

**Summary Of Contributions:**

The submission introduces SpikingBrain, a family of LLMs designed to address efficiency bottlenecks in traditional Transformers, particularly regarding long-context processing and deployment on power-constrained hardware. The authors propose a "conversion-based" training pipeline that adapts a pre-trained Transformer (Qwen2.5-7B) into a brain-inspired architecture.

Strengths
- The conversion-based training pipeline, even though not entirely novel, aligns with the trend of recent training recipes, and is a practical path to adapt pretrained full attention models to subquadratic variants rapidly.
- The implementation on a non-NVIDIA hardware platform is interesting. However, most of the techniques presented don't seem novel. It would be great if the authors could provide a clearer specification of which techniques are merely implementations of existing work and which are the innovations presented in this paper. The MetaX GPUs and MACA frameworks are under-discussed, which makes it hard to appreciate the authors' engineering efforts (e.g., in reimplementing operators in MACA).

Weaknesses
- Experimental rigor. Accuracy comparisons are not data-matched, making the takeaways unconvincing. Important evaluations (e.g., recall) are missing.
- Clarity and presentation issues. Table 1's presentation is dangerously misleading. Multiple typos throughout the paper.
- Limited novelty. The current writing doesn't do justice in highlighting the authors' novel contributions, and reads more like a technical report on the implementation efforts in training what is essentially a Hybrid + MoE + spike coding model.

**Additional Comments:**

- I acknowledge that Qwen2.5 uses full attention and therefore is somewhat of an "accuracy bar" for linear/hybrid models -- and I appreciate the authors acknowledging the performance gap in the evaluation section -- but still, the performance gap is significant, especially when compared to other linear/hybrid models [1].
- One other aspect is to add more details on what didn't work during the adaptation on non-NVIDIA hardware. Right now, the report reads as "what ultimately worked", and detailing what didn't would be helpful to the community.


References
1. Waleffe, Roger, et al. "An empirical study of mamba-based language models." arXiv preprint arXiv:2406.07887 (2024).
2. Arora, Simran, et al. "Just read twice: closing the recall gap for recurrent language models." arXiv preprint arXiv:2407.05483 (2024).
3. Allen-Zhu, Zeyuan, and Yuanzhi Li. "Physics of language models: Part 3.1, knowledge storage and extraction." arXiv preprint arXiv:2309.14316 (2023).

**Audience:**

Yes

**Audience Explanation:**

- The choice of integrating subquadratic attention both inter- and intra-layer is well-motivated to address long-context efficiency.
- The conversion-based training pipeline, even though not entirely novel, aligns with the trend of recent training recipes, and is a practical path to adapt pretrained full attention models to subquadratic variants rapidly.
- The paper makes an effort in presenting a thorough evaluation (e.g., stress-testing on 4M context).

**Broader Impact Concerns:**

The work doesn't present notable ethical implications.

**Claims And Evidence:**

No

**Claims Explanation:**

- When comparing model architectures, it's important to ensure a fair comparison by ensuring that evaluations are data-matched [1,3]. However, Table 1 is an inherently unfair comparison. For example, Falcon-Mamba is a pretrained model, whereas SpikingBrain-7B is adapted from an existing model, therefore adopting a completely different training pipeline.
  + Similarly, "...competitive performance with only ∼2% of the training data typical for mainstream LLMs" is also a slightly exaggerated claim, since SpikingBrain's training is converting an existing base LLM, not pre-training from scratch. I recommend that the authors address this framing.
- A known issue with subquadratic models is their relatively weak recall capabilities [1,2], which are not present in this paper. I acknowledge that the authors evaluate on LongBench, but the dataset is more for long-context understanding and not designed particularly to evaluate on recall.
- Subquadratic attention and hybrid models have extensive prior art (e.g., Qwen3-Next, Kimi-Linear, Nemotron-H) that is neither mentioned nor quantitatively compared with.
- Long-context evaluations (e.g., 4M sequence length with 128 GPUs) should include more details on experiment setups (e.g., hardware details, interconnect details).

**Requested Changes:**

- [Critical] Table 1 is dangerously misleading. The bolded numbers make it seem that SpikingBrain-7B achieves the best performance across benchmarks. However, upon close examination, it is inferior on many tasks (e.g., Qwen2.5 is better across all workloads!). I recommend that the authors fix it by bolding the numbers that represent the best performance.
- [Critical] Include additional evaluations on the recall capabilities.
- [Critical] The abstract claims stable training with MFU at expected levels. However, section 5.4 only reports the MFU of 23.4% on the MetaX cluster, with no mention of the "expected MFU" on NVIDIA equivalents. It would be great to add quantitative comparisons.
- [Critical] Writing changes that address my prior concerns.
- [Critical] Address these typos and presentation issues:
  + Typos: There are multiple places where "Hybrid" is spelled as "Hybird".
  + Nit: HellaSwag is abbreviated as "HS", which is uncommon.
- I think the paper would be greatly strengthened if the authors could include data-matched evaluations to make the takeaways more convincing.
- The adaptive spiking mechanism is interesting. However, its associated implementation efforts (e.g., specialized kernels) are under-discussed. It would be great if the authors could expand Section 4 so that it not only discusses the training details on the MetaX cluster.

---

> ### Author Response · Authors · 2026-02-11
> **[1/6] Response to Reviewer WxpP (Weaknesses)**
>
> We sincerely thank the reviewer for the constructive and technically insightful comments, which have helped improve the completeness and rigor of our work. Below, we address each concern in detail and hope to clarify all remaining issues.
>
> >  **Weaknesses 1: Experimental rigor.**
>
> We ensure that all conclusions and key takeaways in the paper are supported by experimental evidence. During the rebuttal period, we made our best effort to complete the additional experiments suggested by the reviewer.
> (1) For the data-matched experimental results, please refer to response to the *Claim Evidence Challenge 1*.
> (2) For the evaluation of recall capability, please see response to the *Claim Evidence Challenge 2*.
>
>
> >  **Weaknesses 2: Clarity and presentation issues.**
>
> We appreciate the reviewer for pointing out this important issue. We will revise the manuscript accordingly to ensure clearer and more effective communication of the content. Please refer to our responses to *Requested Changes 1 and 5* for details.
>
> >  **Weaknesses 3: Limited novelty.**
>
> We thank the reviewer for the valuable feedback regarding the writing and apologize for any confusion caused. We would like to clarify that integrating these three techniques—Hybrid architecture, MoE, and spike coding—at the scale of large language models, while enabling distributed training and inference on non-NVIDIA clusters, and conducting comprehensive evaluations of both capability and efficiency (including 4M-token benchmarking and spike statistics analysis), is **technically non-trivial** for an academic project.
>
> We believe that successfully implementing such a system and releasing it as open-source already constitutes a meaningful contribution to the community. That said, we acknowledge that in the current submission, the presentation of novelty and the emphasis on system-level implementation may be intertwined. **We will revise the manuscript to more clearly highlight our core contributions**.
>
> We fully acknowledge that novelty is an important consideration in academic research, espescially in the AI/ML literature, which can because it can stimulate new lines of inquiry, challenge established assumptions, and motivate the development of improved methods and systems. We also appreciate the reveiwer's concern on this point. Referring to the [TMLR acceptance criteria](https://jmlr.org/tmlr/acceptance-criteria.html), the acceptance of a TMLR paper relies primarily on:
> - *Whether claims in the submission supported by convincing evidence?*
> - *Would some TMLR's audience be interested in this paper?*
>
> For the first criterion, we worked to make our evidence more accurate, convincing, and clearly presented by adding more additional and more thorough experiments / ablation studies, as well as correcting and clarifying the manuscript where needed; While for the second criterion, while we recognize that our contributions may be more empirical and system-oriented than algorithmically novel, we believe this paper can still be of interest to several segements of the TMLR audience including researchers and practitioners working on:
> -  Frontier models (LLMs, linear attention, hybird models)
> -  Spiking neural networks (spiking encodings, quantinization)
> -  LLM infrastructure (depolyment on non-NVIDIA GPUs)
>
> We hope these revisions clarify the value of the work to the community.

---

> ### Author Response · Authors · 2026-02-11
> **[2/6] Response to Reviewer WxpP (Claim Evidence Challenges 1)**
>
> >  **Challenges 1 & Requested Changes 6: Data-matched evaluations.**
>
> ### Clarification of Table 1
>
> Thank you for raising this concern. We first clarify the intent of Table 1.
> Based on Table 1, we draw the following conclusions (see Section 5.1, Pre-trained Models):
> 1. SpikingBrain-7B recovers nearly 90% of the base model’s performance.
> 2. SpikingBrain-7B achieves performance comparable to advanced Transformer-based models.
> 3. With appropriate training strategies, linear attention architectures can preserve strong general-purpose modeling capability.
> 4. A performance gap remains compared with Qwen2.5-7B, due to substantial architectural modifications.
>
> We believe that **Table 1 sufficiently supports these conclusions**, even though the included open-source models are not strictly data-matched. Importantly, we explicitly report the training data scale for all models in the “Tokens” column.
>
> **The inclusion of Falcon-Mamba results is intended to better support conclusion (3)**: starting from a strong open-source base model and applying a limited amount of data for conversion can outperform same-size models trained from scratch with substantially larger pre-training corpora.
>
> ### Additional Data-Matched Experiments
>
> Although our work does not aim to introduce a fundamentally new architecture—rigorous comparisons among GLA, Transformer, and Mamba have already been established in prior literature—we fully agree with the reviewer that **data-matched evaluations are essential for drawing reliable takeaways**.
>
> For more controlled and strict evidence, **we conducted strict data-matched conversion ablation experiments**, using CPT from Qwen3-1.7B with 5B tokens.
>
> **(1) Attention comparison under identical data budgets**
> We compare the inter-layer hybrid attention paradigm used in SpikingBrain-7B (LA:SWA = 1:1), pure linear attention (LA, similar to Falcon-Mamba), and pure sliding-window attention (SWA, similar to Mistral) under identical conversion settings.
> For SWA training, we use a 4k window size, and report the equivalent window size for all models. For fair comparison, we additionally evaluate a pure SWA variant with the window size fixed to 128.
>
> | Models   | equivalent window size |   HellaSwag ($\uparrow$) |  MMLU ($\uparrow$) | SNIAH1-4k ($\uparrow$) |
> | -------- | ----: | ----: | -------: | --------: |
> | FA Baseline | ∞  | 60.21 | 61.78 |   100.0 |
> | Pure SWA    |4096| 60.12 | 61.74 |   100.0 |
> | Pure SWA    | 128| -     | -     |   1.87  |
> | Pure LA     | 64 | 54.72 | 39.87 |   36.16 |
> | Hybrid      |2080| 57.88 | 56.08 |   96.83 |
>
> **Key takeaways:**
> 1.  **General modeling performance is primarily determined by the degree of architectural deviation between the target and base models.**
> On HellaSwag and MMLU, pure SWA outperforms Hybrid and pure LA, indicating that under data-limited settings, smaller architectural changes enable faster recovery of language modeling capability.
> 2. **Recall performance is primarily governed by effective state size, and LA exhibits stronger recall than SWA with comparable state sizes.**
> Due to faster capability recovery, SWA-4k achieves better recall within its attention window. Although LA requires more tokens to fully recover overall performance, it already demonstrates substantially stronger recall than SWA-128, despite having a similar effective state size.
> 3. **Hybrid attention provides the best overall trade-off.**
> By combining SWA and LA, hybrid attention achieves a balanced combination of reasonable recovery speed, strong recall capability, and reduced memory cost, making it the most effective architecture choice under efficient conversion settings.
>
> **(2) Isolated Impact of H, M, and S Components**
> Under the same experimental setup, we further evaluate the isolated impact of hybrid attention (H), MoE (M), and spike coding (S) during conversion. Details are provided in our response to *Reviewer gn98, Requested Changes 1*. The conclusions are summarized below.
>
> **Key takeaways:**
> 1.  **Hybrid efficient attention (H)** is the dominant contributor to long-context efficiency gains, albeit with some performance degradation.
> 2.  **MoE modules (M)** improve model performance with minimal impact on efficiency.
> 3.  **Spike coding (S)** consistently incurs only limited performance degradation across configurations, while offering potential energy-efficiency benefits under appropriate hardware assumptions.
>
> ### Clarification of "2% of the Training Data"
>
> Thank you for pointing out this ambiguity. We will revise the wording to clearly state that **“2% of the data” refers to the data required for CPT-based conversion, in comparison to training from scratch**.
>
> We have also carefully reviewed related statements in the Abstract, Introduction, Figure 1, Section 3, and other relevant parts of the manuscript, to ensure that readers clearly understand that our emphasis is on the efficiency advantage of the conversion pipeline over pre-training from scratch.

---

> ### Author Response · Authors · 2026-02-11
> **[3/6] Response to Reviewer WxpP (Claim Evidence Challenges 2~3)**
>
> >  **Challenges 2 & Requested Changes 2: Evaluation of recall capability.**
>
> Thank you for raising this concern. In the original manuscript, we evaluated and discussed **long-context modeling and understanding capabilities** (as reported in Section 5.1), which are not designed for recall-intensive scenarios.
>
> We agree with the reviewer that recall is a critical issue for linear and subquadratic models. To answer this, we **conducted additional data-matched conversion experiments** and will **add a dedicated paragraph with relevant citations** in the revised manuscript to explicitly discuss recall behavior.
>
> **(1) Experiments:**
>
> Following the setup described in our response to Challenges 1, we compare **Hybrid, pure SWA, and pure LA** architectures under identical CPT data budgets, and evaluate their **Single-NIAH** performance using the OpenCompass framework on RULER tasks. We additionally include:
> (i) CPT results of a full-attention (FA) Transformer, and
> (ii) an inference-only variant where SWA layers in the Hybrid model are replaced with FA.
>
> | Models   | Equivalent Window Size | SNIAH1-4k / 8k ($\uparrow$) | SNIAH2-4k / 8k ($\uparrow$) | SNIAH3-4k / 8k ($\uparrow$) |
> | -------- | ----: | ----: |  ----: |  ----: |
> | FA Baseline       | ∞  |   100.0 / 98.41 |   98.41 / 96.83 |   96.83 / 95.24 |
> | Pure SWA          |4096|   100.0 / 47.60 |   97.80 / 47.00 |   98.60 / 50.20  |
> | Pure SWA          | 128|   1.87 / 0.00   |   0.93 / 0.93   | 1.87 / 1.87  |
> | Pure LA           | 64 |   36.16 / 39.40 |   11.43 / 12.40 |   0.7 / 0.6 |
> | Hybrid (SWA + LA) |2080|   96.83 / 44.44 |   93.65 / 39.68 |  79.37 / 26.98 |
> | Hybrid (FA + LA)  | ∞  |   74.61 / 71.00 |   92.86 / 55.00 |   97.62 / 31.40 |
>
> **Table Note**: For `96.83 / 44.44`, the first number `96.83` denotes model's accuracy on the SNIAH task with 4k context length, and the second number `44.44` denotes that of 8k context length
>
> **Our observations are as follows:**
> 1. All converted models (SWA, LA, and Hybrid) exhibit degraded recall performance compared to the original FA model.
> 2. A clear trade-off exists between effective state size and recall performance, with linear attention exhibiting stronger recall than SWA at comparable state sizes.
> 3. Hybrid attention provides a more balanced combination of recall capability and state size compared with pure SWA and pure LA.
> 4. Even when trained with SWA and LA, switching to a Hybrid of FA and LA at inference time (similar to SpikingBrain-76B) significantly reduces the long-context recall gap relative to FA.
>
> **(2) More Discussions:**
>
> We further emphasize that **this gap can be further reduced by improving data quality**, for example by incorporating real long-context or recall-intensive training data. Downstream recall performance depends critically on both the quantity and quality of training data. The base Qwen3 model is trained with a 32k context length and includes rich retrieval-style data, whereas our current conversion experiments lack targeted data designed to explicitly elicit recall capabilities in subquadratic models.
>
> Nevertheless, prior industrial studies have shown that **hybrid attention architectures do not significantly impair retrieval performance** relative to Transformers when trained with sufficient private data [1–2].
>
> ---
>
> [1] MiniMax-01: Scaling Foundation Models with Lightning Attention, 2025.
> [2] Qwen3-Next: Towards Ultimate Training & Inference Efficiency, 2025.
>
> >  **Challenges 3: Other hybrid models.**
>
> Thank you for the suggestion. We will **add appropriate references** to acknowledge the strongest hybrid models currently developed in industry.
>
> We would also like to respectfully clarify that, in the academic setting, implementing models of this scale and sequence length on open-source frameworks, particularly adapting distributed training to MetaX clusters, requires **a substantial amount of time and effort**. In fact, this project was initiated in **late 2024**, shortly after the release of Qwen2.5. As such, the central claim of SpikingBrain has consistently been that it nearly closes the performance gap with the base model Qwen2.5-7B, while simultaneously delivering long-context efficiency and potential event-driven energy efficiency.
>
> **We fully acknowledge that many strong open-source hybrid models have emerged over the past year, and we are encouraged to see growing industrial interest in linear attention architectures**. However, it is also important to note that, over the past years, the iteration speed of industrial large-model development especially at leading companies, has significantly outpaced that of academic research. As a result, it is difficult for us to conduct direct quantitative comparisons with very recent models such as Qwen3-Next or Kimi-Linear. Nevertheless, **we believe that the key findings validated in this work remain valuable to the research community, independent of the rapid pace of industrial model releases**.

---

> ### Author Response · Authors · 2026-02-11
> **[4/6] Response to Reviewer WxpP (Claim Evidence Challenges 4 and Requested Changes 1~4)**
>
> >  **Challenges 4: Experiment setups of long-context evaluations.**
>
> We thank the reviewer for the careful examination of our experimental setups. To ensure transparency and reproducibility, we provide detailed hardware configurations and interconnect information for all long-context benchmarking experiments.
>
> 1. 7B model TTFT comparison under SP up to 4M tokens.
> These experiments were conducted on **NVIDIA H100 80GB GPUs**. This hardware platform was selected because support for the ZeCO operator within the MetaX hardware ecosystem is still under active development.
> Within each GPU node, devices are interconnected via **NVIDIA 4th-generation NVLink**, while inter-node communication is enabled through **InfiniBand (400G)**.
>
> 2. 7B and 76B model TTFT comparison without SP up to 128k tokens.
> These experiments were conducted on **MetaX C550 GPUs**. We adapted both SpikingBrain models to the HuggingFace and vLLM frameworks within the MetaX environment.
> Within each GPU node, devices are connected via **MetaX MetaLink**, and inter-node communication is provided by **InfiniBand (400G)**.
>
> 3. 7B model training TGS comparison under SP.
> The hardware configuration is identical to that in (2), and experiments were also conducted on **MetaX C550 GPUs**.
>
> Regarding the hardware used for the 4M-token benchmarking, this information was originally stated in the caption of Table 6 (later moved to the Appendix). **We will ensure that all such hardware and configuration details are explicitly documented in the main text of the revised manuscript**.
>
>
> >  **Requested Changes 1: Table 1 is dangerously misleading.**
>
> We apologize for any confusion. In Tables 1, bolded numbers is used solely to highlight the results of the SpikingBrain models and to distinguish them from the baselines. To avoid ambiguity, we will replace bolded numbers with a gray background for SpikingBrain entries in the revised manuscript. Additionally, as the reviewer suggested, we use the boldface font for the best performing model (which is mostly Qwen-2.5).
>
> >  **Requested Changes 2: Evaluation of recall capability.**
>
> please see our response to the *Claim Evidence Challenge 2*.
>
> >  **Requested Changes 3: Quantitative comparisons of MFU.**
>
> Thank you for the rigorous suggestion. In the abstract, our reference to “expected levels” was intended to indicate that achieving MFU above 20% in multi-node Megatron training is generally considered an acceptable practical standard. We will revise the wording to avoid any potential misunderstanding.
>
> Regarding the requested quantitative comparisons, we conducted the following additional experiments:
>
> 1. **MFU evaluation on NVIDIA A800 80GB under the same parallel strategy.**
> Under identical parallel configurations, the measured MFU on **NVIDIA A800 80GB** reaches **25.8% (TGS 1913)**, which is slightly higher than the result on **MetaX C550**, i.e., **23.4% (TGS 1558)**. This indicates that the NVIDIA ecosystem currently benefits from more mature optimization, particularly in communication and kernel-level efficiency, while there remains room for further optimization in the MetaX stack.
> However, this comparison is not strictly apples-to-apples. Due to the current state of Megatron support on MetaX hardware, we performed substantial interface adaptations and relied on **older software versions (torch 2.1.2 and triton 2.1.0)**, which likely contributed to suboptimal MFU. In contrast, on NVIDIA A800, we followed the recommended Megatron setup and used **newer versions (torch 2.5.0 and triton 3.0.0)**.
>
> 2. **Overall model speed and operator-level comparison on MetaX C550 and NVIDIA A100.**
> We further compared SpikingBrain-7B on MetaX C550 and NVIDIA A100, including both **end-to-end model speed** and **operator-level performance**, while ensuring a fully fair comparison covering environment, model, and framework. Details can be found in our response to *Reviewer gn98, Requested Changes 4*.
> Overall, we verify that **SpikingBrain runs efficiently on both NVIDIA A100 and MetaX C550** using the same implementation at the operator level, and the **end-to-end latency remains within ±2%** across a wide range of tensor shapes. We also observe that **linear-attention kernel optimization** represents the most direct path to further narrowing the remaining overall performance gap.
>
>
> >  **Requested Changes 4: Writing changes.**
>
> We will revise the manuscript accordingly to address the suggested writing improvements.

---

> ### Author Response · Authors · 2026-02-11
> **[5/6] Response to Reviewer WxpP (Requested Changes 5~7)**
>
> >  **Requested Changes 5: Typos and presentation issues.**
>
> 1. Typos: Thank you for pointing this out. Some typographical errors were introduced during figure refinement, and we will correct them accordingly.
> 2. Nit: We followed abbreviations used in some prior works (“HS” for HellaSwag) [1–2]. Based on the reviewer’s suggestion, we will use the full name HellaSwag consistently in the revised manuscript.
>
> ---
>
> [1] Hierarchically Gated Recurrent Neural Network for Sequence Modeling
> , 2023.
> [2] MetaLA: Unified Optimal Linear Approximation to Softmax Attention Map, 2024.
>
> >  **Requested Changes 6: Data-matched evaluations.**
>
> please see our response to the *Claim Evidence Challenge 1*.
>
> >  **Requested Changes 7: Implementation efforts of adaptive spiking mechanism.**
>
> Thank you for your interest in our proposed adaptive spiking mechanism. We first briefly recap the mechanism and then clarify the corresponding implementation efforts under synchronous GPU-based computation and asynchronous neuromorphic execution paradigms. **We will incorporate this discussion into the revised manuscript**.
>
> As described in Section 3.3, we propose a spiking scheme that converts continuous-valued activations into two equivalent representations: **integer spike counts** and **expanded sparse spike trains**. The integer formulation enables efficient execution on GPUs, while the spiking formulation naturally supports event-driven execution.
>
> If the mechanism is used purely for numerical simulation, no special implementation effort is required; however, such usage only serves algorithmic validation and does not yield practical acceleration. To achieve actual performance and energy benefits across different platforms, the following implementation considerations are necessary.
>
> **(1) GPU-based implementation**
>
> On GPUs, acceleration can be achieved by leveraging **INT-type quantized kernels**. In SpikingBrain, this is enabled by `INT8` quantization of weights (Section 5.5), allowing spike-count-based integer operations to be executed efficiently using existing GPU kernel optimizations.
>
> **(2) Asynchronous neuromorphic hardware implementation**
>
> Achieving memory-access and computation energy efficiency on asynchronous hardware requires more specialized, platform-aware design:
>
> - **Algorithm–hardware co-design for memory access**: Conduct offline profiling to identify sparsity pattern of SpikingBrain. On the software side, algorithm-level weight rearrangement and lightweight compression (e.g., run-length encoding) could be performed; on the hardware side, a channel-aware memory controller could be designed and channel-level memory partitioning could be implemented to isolate weights of active and inactive channels into separate memory blocks, enabling effective skipping of inactive channels.
>
> - **Low-overhead asynchronous control**: Adopt lightweight asynchronous protocols to reduce control overhead. For instance, click controllers and two-phase handshake protocols can minimize unnecessary signal switching. Additionally, an adaptive scheduler could be implemented to monitor spike sparsity in real time. The scheduler dynamically decides whether to merge low-activity channels or expand computing resources, further optimizing energy consumption.
>
> - **More comprehensive comparison and end-to-end modeling**: In hardware development, the theoretical estimation model is gradually revised by incorporating previously omitted overheads derived from hardware simulation into the model. The revised model can be used to guide hardware design and algorithm optimization, ensuring that the balance of power, performance and area is achieved, energy efficiency claims are consistent with practical implementation results, and a complete estimation of end-to-end energy consumption is realized.

---

> ### Author Response · Authors · 2026-02-11
> **[6/6] Response to Reviewer WxpP (Additional Comments)**
>
> >  **Additional Comments 1: Performance gap.**
>
> Thank you for raising this important point. We would like to respectfully clarify that training from scratch and conversion training represent **two fundamentally different comparison paradigms** when evaluating performance against Transformer baselines.
>
> In the referenced studies, Transformers and hybrid models are typically **trained from scratch under identical data budgets**, often using trillions of tokens for PT. In this setting, all models benefit from extensive training to fully optimize their modeling capacity. As a result, when a sufficient number of full-attention layers are retained, hybrid models generally achieve language modeling performance (e.g., MMLU) **comparable to, or even exceeding**, that of Transformers.
>
> In contrast, when **converting from an already trained and annealed Transformer checkpoint**, a natural performance gap is expected. The model must adapt to a modified architecture within a **limited additional training window**, which inevitably leads to relative under-training compared to the base Transformer. Consequently, a substantial body of prior work has focused on reducing this gap. For example, when converting from a Transformer to a pure linear-attention model, recovery on MMLU is often limited to around **60%** [1-2]. Other studies show that combining careful distillation with partial full-attention retention can improve recovery to approximately **80%** [3-4].
>
> Notably, **SpikingBrain does not employ any distillation techniques, yet it recovers approximately 89% of the MMLU performance of the base Transformer**. We therefore consider the remaining performance gap to be **acceptable** under the conversion-training paradigm.
>
> ---
> [1] Finetuning Pretrained Transformers into RNNs, 2021.
> [2] Gated Slot Attention for Efficient Linear-Time Sequence Modeling, 2024.
> [3] The Mamba in the Llama: Distilling and Accelerating Hybrid Models, 2024.
> [4] LoLCATs: On Low-Rank Linearizing of Large Language Models, 2024.
>
> >  **Additional Comments 2: What didn't work.**
>
> Thank you for pointing this out. We will add a dedicated section in the revised manuscript to share practical insights about what did not work during the adaptation to MetaX hardware, including:
> - Low GPU utilization, and occasional out-of-memory (OOM) errors and deadlocks during multi-node sequence-parallel (SP) training.
> - The need to adapt Megatron framework interfaces to older versions of PyTorch and Triton, which limited access to newer optimization features.
> - Simplifications in Triton kernel design, such as reduced block sizes and constrained autotuning configurations, which were necessary for stability but resulted in suboptimal performance.
>
> We believe that transparently documenting these limitations and failed attempts will provide useful guidance for future work on deploying large-scale hybrid and spiking models (especially on non-NVIDIA hardware platforms).

---

### Decision · Action_Editor_HS9y · 2026-03-08

**Recommendation:** Accept as is

**Audience:**

Yes

**Audience Explanation:**

This paper discusses an interesting new language model design, which would be interesting for the TMLR's audience.

**Claims And Evidence:**

Yes

**Claims Explanation:**

The main claim about the proposed methodology has been clearly justified by the experiments.